

# Radiometric flight results from the HyperSpectral Imager for Climate Science (HySICS)

Greg Kopp[1], Paul Smith[1], Chris Belting[1], Zach Castleman[1], Ginger Drake[1], Joey Espejo[1], Karl Heuerman[1], James Lanzi[2], and David Stuchlik[2]

[1]Laboratory for Atmospheric and Space Physics, University of Colorado, Boulder, CO 80303, USA
[2]NASA Wallops Flight Facility, Wallops Island, VA 23337

*Correspondence to*: G. Kopp (Greg.Kopp@LASP.Colorado.edu)

**Abstract.** Long-term monitoring of the Earth-reflected solar-spectrum is necessary for discerning and attributing changes in climate. High radiometric-accuracy enables such monitoring over decadal timescales with non-overlapping instruments, and high
precision enables trend detection on shorter timescales. The Hyperspectral Imager for Climate Science (HySICS) is a visible and near-infrared spatial/spectral imaging-spectrometer intended to ultimately achieve ~0.2 % radiometric accuracies of Earth scenes from space, providing an order-of-magnitude improvement over existing space-based imagers. On-orbit calibrations from measurements of spectral solar irradiances acquired by direct views of the Sun enable radiometric calibrations with superior long-term stability than currently possible with any manmade spaceflight light-source or detector. Solar- and lunar-observations enable
in-flight focal-plane-array flat-fielding and other instrument calibrations. The HySICS has demonstrated this solar cross-calibration technique for future spaceflight instrumentation via two high-altitude balloon flights. The second of these two flights acquired high radiometric-accuracy measurements of the ground, clouds, the Earth's limb, and the Moon. Those results and the details of the uncertainty analyses of those flight data are described.

## 1    Introduction

The 2007 NRC Decadal Survey for Earth Science [12] calls for shortwave spatial/spectral Earth-scene measurements with radiometric accuracy and SI-traceability to better than 0.2 % for Earth-climate studies on decadal timescales. These accuracies, being nearly ten times better than current on-orbit capabilities, will establish benchmark measurements of solar radiation scattered by the Earth, provide reference calibrations for other on-orbit instruments, and initiate a climate data-record to be used for future climate-policy decisions.

Current space-based imaging systems have radiometric uncertainties of ~2 % or greater and are limited by the accuracies and stabilities of spaceflight calibration-lamps, atmospheric-correction uncertainties needed for vicarious ground-scene calibrations, and degradation of solar diffusers used for on-orbit instrument-sensitivity tracking. Three prominent and long-duration Earth-imaging NASA instruments, the Moderate Resolution Imaging Spectroradiometer (MODIS), the Sea-viewing Wide Field-of-view Sensor (SeaWiFS), and the Advanced Very High Resolution Radiometer (AVHRR), have radiometric accuracies for their reflective
solar bands of ~2 % (see [5,21,22,23] on MODIS and [1,2] on SeaWiFS) and only cover discrete spectral bands. The NPOESS Preparatory Program Visible/Infrared Imager Radiometer Suite (VIIRS) has similar discrete-band coverage as MODIS with slightly better radiometric accuracies of 1.2 to 1.6 % [24]. Hyperion [13], with continuous spectral-coverage from 400 to 2500 nm and 10-nm spectral resolution, has 3.5 % radiometric uncertainty [3]. With similar spectral coverage and resolution, the Airborne Visible/Infrared Imaging Spectrometer (AVIRIS) has an uncertainty on the order of 4 % [5]. The M[3] and the hyperspectral Visible



Shortwave Infrared (VSWIR) imaging spectrometer for the Hyperspectral Infrared Imager (HyspIRI) Decadal Survey mission have radiometric uncertainties of 5 % [7].

The HyperSpectral Imager for Climate Science (HySICS) is a prototype instrument to demonstrate a new means of achieving ~0.2 % ($1\sigma$) on-orbit radiometric accuracies. This hyperspectral imager utilizes a solar cross-calibration technique whereby

outgoing Earth-radiances of solar-reflected light are ratio-ed to the incoming spectral solar irradiance (SSI) with < 0.2 % relative uncertainty. Unlike other solar-calibrated instruments that rely on indirect-sunlight measurements from attenuating diffusers, the HySICS acquires direct solar-radiance measurements to achieve reduced uncertainties. This solar cross-calibration approach relies on precisely-known attenuation of the incident solar-radiance by $10^{-4.7}$. Attenuations of this magnitude are achieved using a combination of different-sized apertures, electronically-adjustable detector integration-times, and spectral filters having known

transmissions from in-flight calibrations. This spatial/spectral instrument spans the shortwave spectral-range with a single focal-plane-array for reduced mass, volume, power, and cost of potential future spaceflight instrumentation. Two high-altitude balloon flights from above most of the Earth's atmosphere demonstrated the ability to acquire spatial/spectral ground-scene images that were cross-calibrated using SSI measurements to provide radiometrically-calibrated SI-traceable data-cubes.

In this article we provide an overview of the HySICS instrument and describe the solar cross-calibration approach relying on

precisely-characterized attenuation methods (Sect. 2), summarize the two completed high-altitude balloon flights (Sect. 3), detail the data-analysis methods and estimated uncertainties (Sect. 4), and present resulting data-cubes of Earth ground-scenes and the Moon acquired during Flight 2 (Sect. 5).

## 2    HySICS Instrument

An eventual spaceflight-instrument to achieve the 2007 Decadal Survey's solar-reflected Earth-radiance measurement

requirements needed for climate studies would likely be designed to achieve desired ground-scene characterizations having 0.5-km spatial-resolution and 100-km cross-track field of view (FOV) while spanning the 350- to 2300-nm spectral-range with 6-nm spectral-resolution. Acquiring such measurements from low Earth orbit formed the driving requirements for the HySICS spatial/spectral imager, mandating a 10° FOV and a 0.02° instantaneous FOV (IFOV). The HySICS is based on an Offner imaging-spectrometer incorporating a precision ~$10^{-5}$ attenuation system to enable direct measurements of both the Earth and Sun despite

their greatly-disparate radiances.

### 2.1    Optical System

The optical design of the pushbroom HySICS imaging-spectrometer is representative of state-of-the-art hyperspectral imagers, featuring a four-mirror anastigmat (4MA) telescope followed by an Offner spectrometer. The instrument-performance parameters are shown in Table 1.

A precision NIST-calibrated aperture is the first element in the optical train, precisely determining the collecting area for the light entering the instrument. This front-most aperture location allows the most accurate radiometry by reducing uncertainties in estimates of scatter and diffraction effects, which must be corrected to provide low radiometric-uncertainties. Diffraction from the precision-aperture's knife-edge is well understood theoretically, but scatter is surface-dependent and must be measured for the actual optics. Both have been characterized to reduce uncertainties and correct for light losses at the detector. There are no view-

limiting baffles in front of the aperture as these can cause additional diffractive, scattering, and glint effects that are difficult to model and correct. A six-element rotatable aperture-wheel allows selection of any of the HySICS's six circular apertures. A 20-



mm-diameter aperture is used to acquire sufficient signal for Earth-scene radiances while a 0.5-mm-diameter solar-calibration aperture provides a relative attenuation of $10^{-3.2}$ due to the two apertures' geometric areas. Two of each in addition to a 10-mm and a blank-off aperture provide redundancy and a dark mode. Each of the six aperture locations in this wheel has a separate thermistor to allow corrections for thermal expansion of the aperture area.

Immediately following the aperture wheel is a similar wheel containing attenuation filters. These share a common filter-wheel thermistor. A Hg/Ar pen-ray lamp mounted in one of the filter-wheel positions provides occasional spectral calibrations of the downstream spectrometer. Independent control of the aperture and filter wheels allows any aperture-and-filter combination from the six of each installed.

The compact 4MA telescope following the aperture and filter wheels uses aspherical diamond-turned aluminum mirrors with
electroless-nickel coatings. A protected-aluminum topcoat is magneto-rheological-finish (MRF) post-polished for reduced scatter from each element. The fully-reflective system eliminates the need for chromatic corrections over the HySICS's broad spectral-range. The 4MA mirrors and housing incorporate precision-machined mounting-tabs and alignment-pins for mechanical robustness and low sensitivity to thermal distortions. This telescope is designed to produce a distortion-free image of a spatial scene onto a 0.028-mm-wide slit, providing a slit-width-limited spatial-resolution of 0.02° from its 82.2-mm effective focal-length.

The precision 0.028-mm × 14.40-mm rectangular spectrometer-slit was micro-machined by NIST/Boulder. The Offner-facing surface is coated with carbon nanotubes to reduce back reflections. The absolute slit-width was calibrated by NIST/Gaithersburg, as this parameter and its uncertainties are important when reconstructing disk-integrated solar-irradiances from cross-slit scans of the spatially-resolved Sun.

The Offner spectrometer uses independent primary and tertiary mirrors. The secondary element, a convex reflective 100-ln/mm
ruled-grating, provides spectral dispersion, low scatter, and broadband efficiency from its four regions of smoothly-varying blaze-angle. A baffle enclosure machined from black plastic and a zero-order trap limit stray-light inside the spectrometer's enclosing housing. Optical testing demonstrates a spectral-line full-width at half-maximum (FWHM) of < 36 μm at 633 nm, corresponding to a spectral resolution of 3.7 nm with a spectral scale of 103.31 nm/mm at the Offner's focal-plane-array (FPA) detector. Smile- and keystone-distortions are below measureable limits across the FPA.

A three-region order-sorting filter prevents overlap of different orders of diffraction. Region 1, for wavelengths less than 634 nm, is clear; Region 2 passes wavelengths ≥ 634 nm with a 10-nm FWHM transition-region for blocking 2nd-order diffraction; and Region 3 passes wavelengths ≥ 1188 nm with a 39-nm FWHM transition region for blocking 3rd order. This filter is coated on the side of the substrate facing the FPA and is mounted 2.7 mm from the FPA's front surface to improve cut-off sharpness in the converging Offner beam. The order-sorting filter also serves as the entrance window to the FPA's vacuum enclosure that is needed
to allow cryogenic-temperature operations of the detector.

The 480 × 640-pixel substrate-removed 16-bit HgCdTe Teledyne FPA spans the desired spectral range and meets the majority of needed specifications. The 30-μm pixels closely match the spectrometer slit-width. FPA quantum-efficiency ranges from 0.38 $e^-$/ph at 350 nm to 0.76 $e^-$/ph at 2213 nm with a cutoff-wavelength of 2500 nm. Teledyne reported 16 data-numbers (DNs) of read-noise, a 12 $e^-$/DN gain, and a 692 000 $e^-$ full-well for the delivered device; actual results differed slightly, as described in Sect. 4.1.

An operating-temperature of 150 K is achieved via a cryo-cooler and a vacuum-enclosure surrounding the FPA. A small ion-pump helps maintain vacuum during flight.

All optics as well as the aperture and filter wheels are mounted to a thick aluminum baseplate. Three independently-controlled thermo-electric coolers (TECs) reduce thermal gradients of the near-ambient-temperature optics. The entire instrument is encased in a thick aluminum housing for contamination control and thermal stability during integration and test with the balloon gondola

as well as during flight. A small door opens for flight observations, which are performed at flight altitude ambient pressures of ~3 mTorr. A depolarizing door-mounted entrance window can optionally reduce instrument sensitivity to polarized scenes when the door is closed (albeit at the expense of additional light losses due to the window surfaces).

A separate electronics box contains all the controlling components for the HySICS optical module. This 1-atmosphere nitrogen-

pressurized enclosure is maintained during flight since not all off-the-shelf electronic components are intended for near-vacuum operations. The FPA electronics are mounted in this box in close proximity to the FPA for reduced noise. Five-hundred gigabytes of solid-state memory arranged as a redundant array of independent disks store all data redundantly during flight, allowing up to eight hours of continual, uncompressed, 14-Hz imagery from the FPA.

### 2.2    Solar Attenuation System

Three methods collectively provide the required $10^{-4.7}$ attenuation for directly viewing the Sun: reducing optical-entrance aperture-size, decreasing detector integration-times, and inserting attenuating filters. The specifics of these three attenuation-methods are detailed below. The attenuations collectively provided by the aperture-ratio and the integration-time methods proved sufficient for the needed solar-attenuation range, making the filter-based attenuation method unnecessary; nevertheless, that system was incorporated in the HySICS and flight-validated as well.

#### 2.2.1    Aperture Attenuation-Method

Changing from an entrance-aperture diameter of 20 mm for viewing Earth-scenes to 0.5 mm for viewing the Sun provides a geometric attenuation-level of $10^{-3.2}$. Optical-system complexities disfavor the use of larger apertures while diffraction-loss uncertainties start to preclude the use of significantly smaller ones to obtain greater attenuation-ratios via this method.

The HySICS apertures are diamond-turned nickel-coated aluminum, providing a very sharp aperture-edge with nearly-negligible

scatter. The six installed have entrance diameters of 20, 10, and 0.5 mm, with two each of the largest and smallest. All apertures are calibrated by NIST/Gaithersburg for geometric area using a non-contact optical technique to achieve the desired attenuation-uncertainties, with the limiting factor being the 0.06 to 0.08 % ($1\sigma$) relative area-uncertainties of the solar-viewing 0.5-mm-diameter apertures.

#### 2.2.2    Integration-Time Attenuation-Method

Shorter integration-times are used for solar-viewing than Earth-scene measurements. These are enabled by the FPA electronics, reproducible detector-linearity, and an electronic global shutter to avoid spatial smear during image integration.

The FPA's controlling electronics demonstrate < 14 ns timing-stability and linearity to < $10^{-6}$ over the integration time-range from 16.8 μs to 34.4 ms, providing $10^{-3.3}$ solar-attenuation capability. The FPA response itself, unsurprisingly, has higher non-linearities but nevertheless demonstrates sufficient linearity-stability to allow corrections for operation over the large applied intensity-range.

#### 2.2.3    Filter Attenuation-Method

Spectral filters capable of roughly $10^{-1}$ attenuations can be calibrated on-orbit via lunar observations. Greater filter-based attenuations are precluded by low lunar-radiance levels that would limit the accuracies of these on-orbit calibrations. On-orbit spectral filter calibrations using the Sun are also possible because of the large integration-time range achievable with the FPA.




These solar-based calibrations benefit from the use of the same small aperture (and thus the same optical path) used in operations when acquiring solar observations with the filters.

The three ionically-colored Schott glass filters in the HySICS were polished to 0.1 nm RMS surface-roughness and λ/4 flatness to reduce induced scatter and distortion. The balloon-flight filter selection includes:

1. NG4 (a neutral-density filter with ~0.1 transmission),
2. NG5 (a neutral-density filter with ~0.3 transmission), and
3. BG25 (a high-transmittance filter in the UV and IR).

## 3    High-Altitude Balloon Flights

The HySICS was flown on two high-altitude balloon flights to demonstrate its ability to cross-calibrate Earth-scene radiances to
the spectral solar irradiance. Each of the ~9-hr flights maintained a float altitude of 39 000 m (120 000 ft) to acquire SSI measurements in the near-absence of attenuations or scatter by the Earth's atmosphere.

### 3.1    Balloon-System Design

The HySICS was mounted in a two-axis gimballed pointing system able to track the Sun and Moon for calibrations and able to maintain a fixed-angle nadir view for scanning along the ground as the balloon drifted. The pointing system was mounted near the
center of a large rectangular-frame gondola that was suspended from the balloon itself. A rotator mechanism between the balloon and gondola provided coarse azimuthal-pointing (±3°) of the latter while the gondola-based WASP provided fine-pointing of the instrument itself.

#### 3.1.1    WASP System

The Wallops Arc Second Pointer (WASP) is a two-axis altitude-azimuth gimbal-based pointing-system designed to achieve nearly
arc-second accuracy-levels for balloon payloads [19,18]. This system was provided courtesy of HySICS co-investigators D. Stuchlik's and J. Lanzi's team at NASA's Wallops Flight Facility (WFF). With the HySICS center-of-mass aligned within the WASP gimbal-axes to ±25 μm, the system is able to point the instrument accurately at the ground, Sun, and Moon and track each of these objects while acquiring the needed measurements and calibrations.

The WASP generally provided < 10 arcsec of pointing accuracy during Flight #2, meeting the HySICS pointing requirements. The
most critical pointing-requirements are driven by scanning the Sun or the Moon lengthwise along the HySICS slit to obtain FPA flat-fields by positioning the same portion of the Sun or Moon on each pixel in the FPA's spatial direction. Pointing knowledge and after-the-fact corrections are not sufficient for this flat-fielding calibration-method; real-time pointing accuracy is needed. The WASP system achieved approximately 8-arcsec ($1\sigma$) pointing deviations across the ±6° range about disk-center for along-slit solar-scans, acquiring the needed flat-field calibrations. Accuracies of 2-arcsec ($1\sigma$) across the ±1.5° range for cross-slit solar-scans were
achieved, with these scans intended to acquire solar-irradiance measurements by spatially-integrating sequential images across the solar disk via post-flight ground-based data-processing. The WASP provided 0.7-arcsec ($1\sigma$) along-slit stability and 2.0-arcsec cross-slit stability when staring at the Sun. While the WASP is generally capable of yet more accurate pointing, that provided during flight was sufficient for the HySICS's purposes.



The WASP was also able to inertially-track the Moon using an on-board ephemeris. This new pointing-system capability enabled flat-fielding calibrations using the Moon while operating with the same 20-mm aperture (and thus optical paths) and integration-time parameters as used for Earth-scene observations.

### 3.1.2 Gondola

The balloon gondola is a rectangular-frame structure that houses the entire payload, consisting of the HySICS instrument, the WASP, 27 lead-acid batteries to supply power, all telemetry- and tracking-equipment, thermal enclosures, several crush-pads for landing, and ballast. The net mass of the payload and gondola is 2300 kg (5000 lbs.) including 540 kg (1200 lbs.) of ballast.

The gondola was designed and built at the University of Colorado's Laboratory for Atmospheric and Space Physics (LASP) using a combination of 80-20 aluminum and square aluminum-tubing. The structure is 3 m in height and contained within a 4.3-m

diameter region when the crush pads are installed on all but the top of the gondola's six rectangular sides. During flight the entire structure is suspended by the azimuthal rotator that provides coarse pointing.

The WASP and HySICS are centrally located in the gondola such that the HySICS can view nadir for observing the Earth and greater elevation-angles for solar- and lunar-measurements. Once expanded at altitude, the overhead Helium-filled balloon restricts viewing to elevation angles < 60°. A remotely-controlled caging mechanism locks the WASP to the gondola frame for launch and

parachute-descent landing.

### 3.2 Flight Summaries

Both high-altitude balloon flights were performed out of Fort Sumner, NM and supported by the Columbia Scientific Balloon Facility (CSBF). Upper-atmosphere winds limit Fort Sumner balloon-flights to a few weeks in the springtime and fall, while CSBF schedules limit support at Fort Sumner to only the fall launch season. Ground winds generally limit launches to early mornings.

Upper-atmosphere wind speeds determine flight duration and allow only a narrow timeframe of a couple of weeks for lengthy flights needed for many other programs' nighttime viewing. HySICS observations allow a more extended launch-window since the Sun and Earth are the primary targets and both can be viewed shortly after the morning launches; nighttime observations are not needed.

Lunar observations, however, are needed, as they allow flat-fielding using the same optics as for ground-viewing. While low lunar-

phases are beneficial for the higher-radiances provided near full moon, such nighttime-acquired flat-fields would be separated temporally from the Earth-ground scenes and would also require longer flight durations. Instead, higher lunar-phase angles were chosen to acquire the flat-field calibrations at similar instrument temperatures and times to the acquired ground scenes. Launch windows at less than 90° lunar phase were desired so that likely flight durations would include daytime solar-observations along with early-morning or late-evening lunar-observations. Unfortunately, these were precluded by high ground-winds preventing

launch attempts during the HySICS flight campaigns. Instead, both flights occurred with higher-than-desired lunar-phase. The low lunar-signals due to these high phases limited achieving the desired low uncertainties for flat-fielding and filter calibrations with the Moon. Nevertheless, all intended observations were acquired to demonstrate all aspects of and the achievable capabilities of the HySICS solar cross-calibration methods. (Such lunar-phase restrictions would be alleviated from space-borne platforms having more extended lunar-observing times.)

Flight 1 occurred on 29 Sept. 2013 with launch at 13:30 UT and landing at 22:13 UT. A float altitude of 37 100 m (121 800 ft) was reached for this engineering flight, during which the HySICS and WASP attempted all needed measurements. The gondola was recovered and returned to LASP for refurbishment. No damage to the instrument occurred during this flight or landing. Flight 2



launched at 15:36 UT on 18 Aug. 2014, reached a float altitude of 37 200 m (122 000 ft) at 17:52 UT, was powered off at 23:52 UT, and landed early on the following day. Despite a rough landing, post-recovery checkout revealed that the instrument was unharmed and all optical alignments were maintained, validating the HySICS's robust design.

### 3.3 Flight Observations

The HySICS has three primary observation targets, each containing various observation-modes, as well as several internal-instrument calibrations.

### 3.3.1 Ground Scans

These cross-track scans, with the ground-track and -speed determined by the balloon velocity from the aloft winds, provide samples of the desired data from an eventual flight-instrument. During Flight 2 four ground-scans were acquired. Two in the morning

included a mix of the New Mexico high-desert with broken clouds while the two in the afternoon were predominantly of high, thin clouds. Three-dimensional data-cubes of these scans were created in ground processing after all radiometric calibrations were applied.

Several scans of the Earth-limb were also obtained on this flight. These scans provide spatial-spectral information through the vertical extent of the Earth's atmosphere. The Earth-limb itself was largely occulted by the tops of bright cumulus clouds at the

near-horizontal look-angle for these scans. Some such scans also included the Moon as it was setting.

### 3.3.2 Solar Scans

Along-slit scans enable flat-fielding of the FPA by placing the same portion of the Sun on each spatial element of the array. Cross-slit scans build up an entire data-cube of the Sun, enabling the spatially-integrated solar-irradiance to be determined and allowing SI-traceability to SSI (provided on an absolute scale by other measurements or models), as detailed in Sect. 4.4. Since

demonstrating the solar cross-calibration method was the primary purpose of these flights, solar scans dominated the flight observation time. Near local-noon the Sun's elevation was greater than 60° so solar observations could not be acquired due to glint or occultation by the large overhead balloon. At these times either lunar- or ground-scenes were acquired instead.

### 3.3.3 Lunar Observations

Similar to those done with the Sun, along-slit scans enable flat-fielding of the FPA by placing the same portion of the Moon on

each spatial element of the array. The lunar scans can be done with the larger Earth-viewing aperture, potentially providing a more appropriate flat-field to be applied to ground scans than those obtained from solar scans. Additionally, spectral-filter transmission is calibrated during flight by quick successive measurements with each filter in and out of the optical path while tracking a fixed position of the Moon.

### 3.3.4 Internal-Instrument Calibrations

Internal-instrument calibrations and diagnostics helped track instrument functionality, stability, and performance in-flight. Spectral calibrations were made intermittently throughout the flights by briefly illuminating the instrument's Hg/Ar pen-ray lamp. Pointing stability was quantified by attempting to maintain the instrument slit at a fixed position on the edge of the lunar limb for an extended period. At this position, lunar intensity is very sensitive to cross-slit variations in pointing, providing a diagnostic of pointing



stability. An along-slit scan at this lunar position quantified the instrument's alignment relative to the WASP's elevation (altitude) direction.

## 4    Flight 2 Data Analysis and Uncertainties

The intent of Flight 2 was to quantify the radiometric uncertainties to which HySICS-acquired Earth-scenes could be related to
known spectral solar irradiances. The HySICS spatial/spectral ground-images, $S_{\mathrm{meas\_obj}}(\lambda)$, which are measured in units of instrument DNs, are converted to physical units of spectral solar irradiance (such as W m$^{-2}$ nm$^{-1}$) by applying a scale factor for an on-orbit-determined unit-conversion-factor, $C(\lambda)$ (in units of spectral solar irradiance per DN), and the instrument's unit-less, ground-calibrated radiance-attenuation factor, $A(\lambda)$, which corrects for the optical-throughput and integration-times used for solar-vs. Earth-viewing, according to

$$S_{\mathrm{SI}}(\lambda) = S_{\mathrm{meas\_obj}}(\lambda)\, A(\lambda)\, C(\lambda), \tag{1}$$

where $S_{\mathrm{SI}}(\lambda)$ represents the radiance of the observed scene in SI-traceable, physical units. The unit-conversion factor $C(\lambda)$ has the form

$$C(\lambda) = SSI(\lambda)\, /\, S_{\mathrm{meas\_Sun}}(\lambda) \tag{2}$$

where $SSI(\lambda)$ is the spectral solar irradiance (provided by an independent spaceflight instrument or a solar model), and $S_{\mathrm{meas\_Sun}}(\lambda)$
is the HySICS's in-flight measurement of the SSI in DNs acquired by spatially-integrated cross-slit scans of the solar disk. Equation (1) is thus effectively a ratio of two in-flight HySICS measurements, $S_{\mathrm{meas\_obj}}(\lambda)\, /\, S_{\mathrm{meas\_Sun}}(\lambda)$, and calibration factors, $A(\lambda)$, to account for solar- and Earth-scene attenuations. Being a ratio, accurate on-orbit knowledge of common-mode instrument efficiencies are not critical for acquiring radiometrically-accurate ground measurements. Since the needed solar- and Earth-measurements can be acquired in close temporal sequence, this on-orbit solar cross-calibration method, HySICS's SI-traceable
measurements of ground scenes are not susceptible to potential long-term in-flight degradation of the instrument optics. This method ties the long-term accuracy of the HySICS to the accuracy to which the SSI is known and the long-term stability of the instrument's attenuation systems. These latter two are based on physical components, such as geometric aperture-sizes and electronic timing, such as that controlling detector integration-times; both are inherently very stable.

Since the factors in Eq. (1) are independent, their individual uncertainties are evaluated separately and root-sum-squared for each
final scene-dependent uncertainty. These correction factors and their uncertainties are derived from component- and instrument-level characterizations from both pre-flight laboratory-based calibrations and in-flight calibrations of the instrument, and are described in this section.

### 4.1    Focal-Plane-Array Corrections and Uncertainties

The initial data-analysis step is to apply corrections to the raw data-images. Applying all such corrections gives $S_{\mathrm{meas\_obj}}(\lambda)$ in Eq.
(1). The initial corrections are detector-specific and are typical of any FPA-based instrument, so are only cursorily mentioned in this sub-section for completeness.




### 4.1.1 Bad-Pixel Removal

Non-responsive pixels and badly-fluctuating pixels, defined as those with a measurement-to-measurement standard deviation of more than $5\sigma$ greater than the sensor-wide average standard deviation, are filled using an average of all properly-operating neighboring pixels. The HySICS FPA had 732 pixels needing such corrections. These are sufficiently few that they do not greatly

influence subsequent statistics based on full-FPA data using their corrected values.

### 4.1.2 Read-Noise

Read noise for the Teledyne sensor is determined using a traditional photon-transfer measurement [8] of a constant radiant-power source provided by blackbody radiation from a uniform, warm temperature-stabilized target. This target is measured at various exposure levels by varying the integration time from 33.6 μs to 34.4 ms. A corresponding dark image, acquired during a prior

measurement of a 77 K target to eliminate blackbody radiation, is subtracted from each exposure in the photon-transfer measurement. The measured noise on each pixel, given by the standard deviation of 50 repeated measurements, is dominated by read noise at the shortest integration times and by shot noise at the longest integration times. Although a true zero integration-time cannot be achieved, the noise versus signal level for each pixel is curve fit to an expected photon-transfer curve to extrapolate to its true read noise. The sensor-wide average read noise is 8.3 DN.

With a gain of ~12 e$^-$/DN (see Sect. 4.1.5), read-noise uncertainties are thus based on random fluctuations around 100 e$^-$. Since these are of similar amplitude across the array and are independent of incident signal, read-noise causes a higher relative uncertainty at low signals-levels, such as the extreme portions of the spectral range where the solar signal and the detector response are both low. Higher signal-levels, such as can be achieved from brighter scenes or longer integration-times, reduce the effects of read-noise uncertainties. By acquiring all solar calibrations at both short and long integration-times, read-noise in select wavelength

ranges is greatly improved. Similarly, since read-noise is a random statistical fluctuation, acquiring repeated images of the same scene reduces the effects of read-noise as the reciprocal square-root of the number of images. Such integration-time variations and multiple-image acquisitions are not possible when viewing the ground during flight since balloon-track motion between frames causes either a different ground scene (for static nadir-viewing) or a different look-angle of the same ground scene (if actively tracking) to be measured by non-simultaneous successive frames; however, multiple-image acquisitions are implemented for

HySICS calibrations using static sources such as the Sun and Moon. Thus, read-noise mainly contributes to the ground-measurement uncertainties at shorter wavelengths.

### 4.1.3 Dark and Thermal-Background Corrections

Both dark-noise and thermal-background signals from the surrounding instrument scale with integration time and are dependent on instrument- or FPA-temperature. Thermistors monitor the FPA and several of the nearby instrument-components. Laboratory

characterizations of the dark signal enable corrections for both internal-FPA and background-thermal effects.
The HySICS FPA's inherent dark signal is sufficiently low that it is difficult to detect in the presence of any background light. A cold target placed in front of the imager while keeping the sensor housing at -25 C reduced such background signals but did not completely eliminate them sufficiently. Dark current, which increases with FPA operating-temperature, was therefore measured at elevated operating temperatures of 165 K and warmer. These measurements were extrapolated to the FPA's nominal 150 K

operating-temperature, yielding a dark current of 350 e$^-$/sec and resulting uncertainties of 0.29 DN for the longest HySICS integration times (34.4 msec) used. These inherent dark-signal uncertainties are well below the quantization limit of the device.



Background signals were also corrected during flight. Following all data acquisitions, the blanked aperture wheel position blocked incoming light for 100 exposures. These consist only of dark current, instrument thermal-background contributions, and imager fixed-pattern noise. They are acquired at the same integration time and nearly the same temperatures as the data frames themselves. The average of these dark exposures is subtracted from the data frames, thereby removing background offsets with the exception

of possible thermal offsets caused by temperature differences between when the data and the dark measurements were acquired. These temperature dependencies are in turn corrected via in-flight thermal-background measurements using portions of the array viewing dark space during solar- and lunar-scans. From multiple such scans, FPA sensitivities to instrument thermal effects are determined as a function of surrounding instrument-component temperatures. All raw HySICS data-images are thus corrected for thermal background based on the instrument temperatures at the actual time of data acquisition using the instrument-temperature

dependencies determined from these dark-space observations.

Although these thermal-background signals are largest at the longer-wavelength portion of the FPA's sensitivity, they influence the entire array uniformly since the FPA has no long-wave rejection filter over the portions used only for shorter-wavelength readout, making the above corrections necessary for all portions of the spectrum. While the dark current is very small and contributes nearly insignificantly to the net HySICS uncertainties, the thermal-background signal contributes to shot noise

(described in Sect. 4.2.1).

### 4.1.4 Linearity Corrections

Deviations from linearity are determined individually for each FPA pixel in laboratory testing using varying levels of incident-light intensity and integration times. If temporally stable, non-linearities can be corrected once characterized. These corrections are applied to the images after the bad-pixel, dark, and thermal-background corrections.

Sensor linearity is measured in two steps: 1) The electronically-determined integration-time is measured directly using timing pulses from the sensor's field-programmable gate-array's digital output-signal; and 2) the response of the FPA itself is measured using a stable light-source while varying the now-known electronically-controlled integration-time. The former verifies the timing of the controlling electronics, which are, as expected for oscillator-based signals, very linear and stable. The latter step includes the effects of FPA pixel-well or amplifier-signal saturation and is a function of the net signal on each pixel. To characterize these

non-linearities, the sensor is illuminated by a stable FEL lamp while the electronically-controlled integration-time is varied and the resulting signal-levels are measured. A linear curve-fit is used to determine the expected signal-level on each pixel, and deviations from that fit with signal level are considered non-linearities in that pixel's response. The curve fit uses only the most linear portion of the data at less than 50% of the FPA's full-well. Repetition of this measurement, using various FEL-lamp intensities, ensures that the deviation from linearity has an FPA signal-level dependence rather than an integration-time

dependence.

The resulting non-linearities and uncertainties are detailed in Sect. 4.3.2, where the non-linearity corrections, uncertainties, and intensity range and the resulting dominant determinants of the overall instrument attenuation-uncertainty based on the integration-time method are discussed.

### 4.1.5 Pixel-Dependent Gain Determinations

Sensor gain, or the conversion [e⁻/DN] from FPA DNs to electrons [e⁻] and thus photons, is determined from a photon-transfer measurement in laboratory testing on a pixel-by-pixel basis using statistics of each pixel's variations at different intensity-exposure levels. This was done using the same experimental setup as the read-noise measurement described in Sect. 4.1.2. In the larger-





signal regime where pixel noise is dominated by shot noise, sensor gain is defined as the ratio of signal level to pixel-noise variance. The previously-determined read-noise variance is subtracted from the measured pixel-noise variance so that the residual noise is that due solely to shot noise. For each pixel, the signal level and pixel-noise are determined using 50 exposures repeated at ten different signal levels. The experiment is repeated 100 times to determine the average gain and to reduce the shot-noise

measurement uncertainty. The sensor-wide average pixel-gain is 12.01 e⁻/DN with an average uncertainty per pixel of 0.12 e⁻/DN, or < 0.003 % uncertainty in the shot-noise calculation at a signal level of 15 % (10 000 DN) of full scale. This pixel-dependent correction is applied to each pixel in the array but is an insignificant contributor to the net uncertainties.

### 4.1.6    Flat-Field Corrections

Flat-fielding the HySICS sensor requires a full-system calibration since it is affected by the collective efficiencies of all upstream

optics. This calibration therefore needs to be performed separately for the smaller solar-viewing aperture and the larger Earth-viewing aperture, as light passing through the two apertures interacts with different portions of the downstream optical-elements in the instrument. These differences are accounted for via the flat-field calibrations and are corrected in post-processing of the data. Although different apertures are used for the two scans, the flat-fielding procedure for both is to use a stable light-source that can be swept across every pixel on the sensor, enabling a measurement of the relative response, or gain, of each. In space, the only

available sufficiently-stable light-sources are the Sun and the Moon, which are used for the small- and large-aperture flat-field calibrations respectively. In both cases, a slice near the center of the solar- or lunar-disk is scanned in the along-slit direction from one edge of the imager to the other while images are captured at the instrument's nominal 14-Hz cadence used for ground-scene measurements. The flat-field calibration scans ±6.5° from the boresight, going 1.5° outside the HySICS's FOV in both directions to ensure full spatial coverage. At a scan rate of 5.88 arcmin/sec, the along-slit flat-field scan requires 102 seconds

to complete, during which time both sources are considered stable even during high rates of change in lunar phase. (Diffraction from the small solar-viewing aperture spatially blurs the Sun so that fine detail due to solar oscillations or granulation, which vary on 5- to 10-minute timescales, are not observable and sensitivity to pointing errors is small. The lunar flat-field scan, however, is more sensitive to pointing and slit-alignment errors because of large intensity-variations across the lunar crescent and – unlike viewing the Sun through the small aperture – there is very little diffraction to spatially-blur the image.) The resulting data are a

series of images containing the solar- or lunar-spectrum stretched across the full sensor in the spectral direction and gradually moving through the sensor's entire spatial direction with each successive image. This technique essentially sweeps an identical spectrum across the spatial-direction of the array, by which the spatial-direction flat-fielding is accomplished. (The spectral-direction flat-fielding is done at a later data-processing stage when the HySICS measurements are calibrated to the independently-known SSI.) Since both sources extend over multiple spatial pixels, the pixel-to-pixel signal comparison can be repeated multiple

times and utilized in measurement averaging as well as providing a basis for uncertainty estimations. A total of 31 spatial positions across the solar disk are applied to the flat-field correction, while only 9 positions across the narrow lunar crescent during Flight 2 are used. Uncertainties for select wavelengths are summarized in Table 2.

As with read-noise, since the flat-field calibrations are acquired using static sources, they can benefit from multi-acquisition scans to reduce random uncertainties and at different integration times to improve signal in spectral regions having lower sensitivity such

as the visible. These approaches, described in more detail in Sect. 4.2.6, were not performed for the flat-field calibrations of either the Sun or the Moon during Flight 2, and as a result the flight-acquired flat-field uncertainties dominate all others at the shorter wavelengths where instrument sensitivity is low. For flat-field calibrations using the Sun, cross-slit scans of which did benefit from multi-acquisition scans at different integration times and thus have low uncertainties for most other parameters, the flat-field



uncertainties dominate at the shorter wavelengths and are comparable to diffraction at the longer wavelengths, so would greatly benefit from multi-acquisition scans. Lunar flat-field calibrations were marginal because of the high lunar-phase during the time of the flight, giving low lunar signal and small spatial extent. These along-slit lunar scans are not only low in signal, but very sensitive to pointing, particularly since the large aperture used for lunar flat-field calibrations does not spatially blur the lunar

image due to diffraction as the smaller aperture does to the solar image. Where the acquired flat-field uncertainties exceed the array's intrinsic 3.3 % pixel-to-pixel variations, such as in the shorter-wavelength portion of the visible, they were clipped at this intrinsic value.

Multi-acquisition flat-field calibrations at different integration times for both the Sun and the Moon and a lower lunar phase-angle would greatly improve the uncertainties demonstrated by Flight 2. Nevertheless, in spectral regions having high signal, the flat-

field uncertainties acquired during this flight are < 0.2 %, demonstrating the flat-field calibration method capabilities and showing promise of achieving desired lower uncertainties across a broader spectral range with the suggested multi-acquisition approach. Flat-field corrections are applied to ground scenes and cross-slit solar scans and thus these uncertainties directly affect those data. Measurements that rely purely on relative measurements, such as calibrations of aperture-ratio (Sect. 4.3.1) and filter-transmission (Sect. 4.3.3), are not affected by these flat-field uncertainties.

**4.2    Instrument Uncertainties**

Sect. 4.1 discussed corrections from the FPA and their associated uncertainties. Further contributions to $S_{\mathrm{meas\_obj}}(\lambda)$ in Eq. (1) account for higher-level aspects of the HySICS's optical performance, which was evaluated at both component- and integrated-levels. The component-level tests validated or refined modeled performance at each stage during assembly. The tests indicated expected performance for most components, including the 4MA, apertures, filters, slit, and Offner mirrors. High-level

uncertainties, such as those from photon counting, diffraction, optical scatter, varying optical paths, opto-mechanical or thermal effects on spectral scale, and polarization sensitivity are described in the following sub-sections.

**4.2.1    Shot-Noise**

Shot-noise arises from photon-counting statistics and varies as the reciprocal square-root of the signal, including that from any thermal background. As with read-noise, it is reduced via multiple-image acquisitions for calibrations of static sources, namely the

Sun and the Moon, but cannot be similarly reduced for single-acquisition images of the ground. Shot-noise is the dominant source of uncertainty for ground-scenes across the majority of the spectrum.

**4.2.2    Diffraction**

HySICS directly measures outgoing Earth-reflected shortwave radiances and incoming solar radiances. By spatially-integrating radiances from the entire solar disk, which are acquired from cross-slit scans of the Sun, the HySICS measurements are calibrated

to the independently-known incoming SSI. To provide an accurate spatial integration, HySICS data analysis needs to correct for radiative losses, such as due to stray-light and diffraction, that may cause differences in the amount of light reaching the FPA when viewing the Sun as opposed to ground scenes. Losses from diffraction are higher for the solar-viewing configuration than for ground-viewing because of the smaller aperture used for solar observations. (Figure 1 illustrates the noticeably-larger diffraction that must be accounted for when using the 0.5-mm solar aperture compared to the 20-mm Earth-viewing aperture. At 1000 nm, the

diffraction limit from each is ~8 and 0.2 arcmin, respectively.) Spatial scans sweeping across and then well away from the Sun



provide scatter- and diffraction-characterizations. These, in addition to lab measurements of the same, enable corrections to facilitate accurate determinations of the net SSI based on cross-slit scans of the Sun.

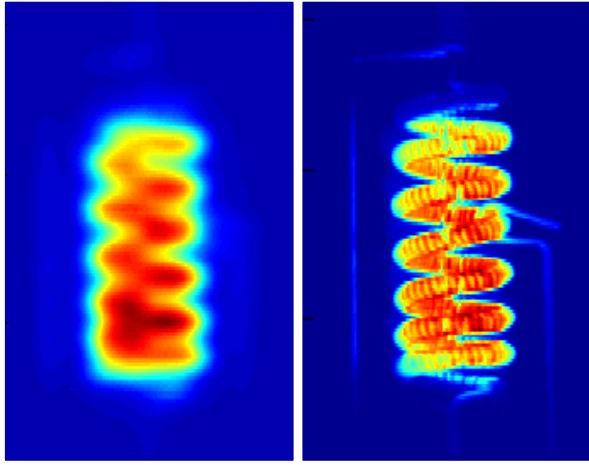

Figure 1: These scans of a lab FEL-lamp filament show the burring caused by diffraction when using the 0.5-mm aperture
5   (left-hand image) vs. the 20-mm aperture (right-hand image). These effects must be accounted for in spatially-integrated spectral solar irradiance determinations.

Diffraction can be modeled well with a NIST-quoted uncertainty of ~1.8 % for simple circular-aperture geometries [16]. The effects of scatter, however, are very instrument-specific and can be more difficult to model in advance to sufficient levels of accuracy. Lab measurements using the setup shown in Figure 2 helped determine these contributions by characterizing the HySICS

10   system's diffraction and scatter properties. This experiment occults the light coming directly through the aperture but captures most of the light scattered or diffracted from it, then re-images that light onto a separate FPA. Sample results are shown in Figure 3 and match the expected angular dependence due to diffraction alone, indicating that diffraction, as opposed to scatter, is the dominant source of this indirect light for the as-built HySICS.




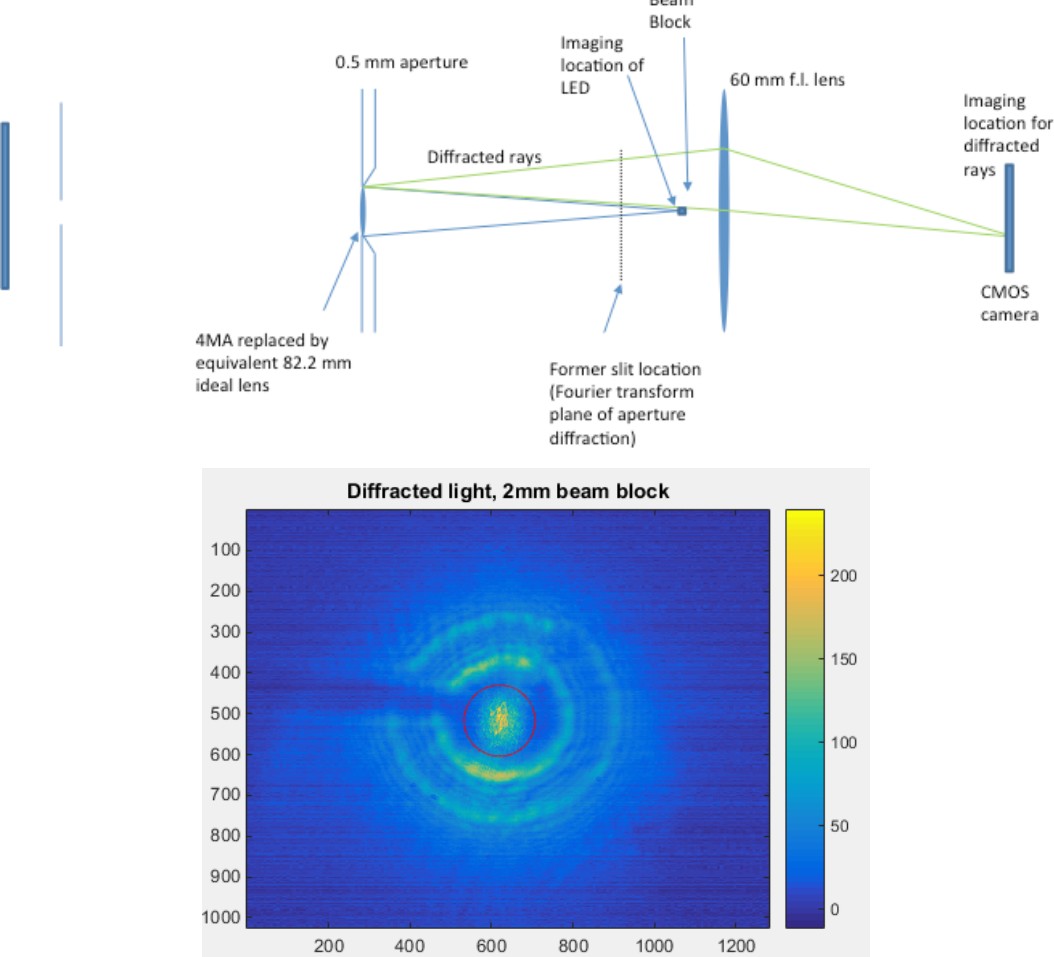

Figure 2: Lab scatter- and diffraction-characterization setup (upper plot) gives the 2-D pattern shown in the lower image when using a 2-mm beam block behind the 0.5-mm aperture. These effects must be corrected when determining net solar-irradiance values via spatial integrations from cross-slit scans of the solar disk. (The nearly-horizontal radially-extending dark region to the left of image center is due to the support for the beam block.)



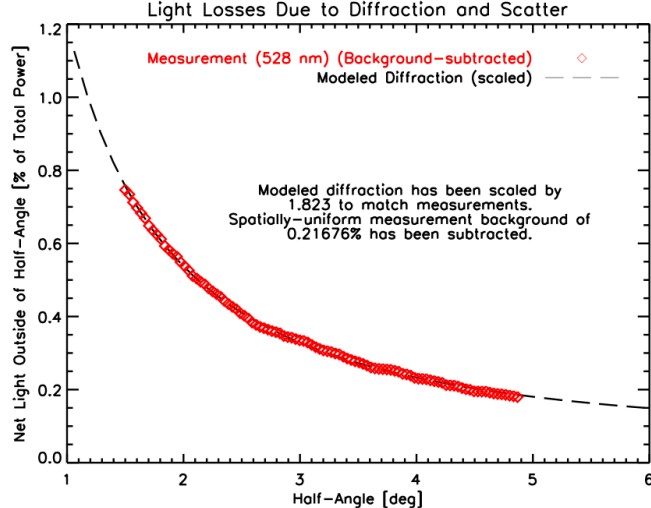

Figure 3: An idealized model estimating the net amount of light falling outside of various angles due to diffraction alone (black dashed curve) is scaled to match lab measurements including both scatter and diffraction (red diamonds) at 528 nm. The measurements match the angular dependence expected from diffraction, indicating that the majority of this measured light loss is mainly due to diffraction rather than scatter, which would have a less-well-modeled relation to angle. Correcting for light losses via this validated diffraction-model reduces the uncertainties in solar-radiance measurements, improving the HySICS's determination of spectral solar irradiance.

By modeling the diffracted light and verifying that model with lab measurements, the expected light-losses are accounted for when spatially-integrating the solar disk to obtain a value that can be correctly calibrated to the independently-known SSI. A 1.8 % uncertainty on this correction is allocated per NIST diffraction-estimate uncertainties. Because diffraction scales with wavelength, these corrections begin to dominate the solar-calibration uncertainties at the longer wavelengths but never greatly exceed the contributions from read- and shot-noise. The HySICS's small solar-viewing aperture was chosen such that uncertainties due to diffraction may be the limiting uncertainty at the longest wavelengths but would not dominate across the spectrum, effectively balancing desirable greater-attenuation capabilities afforded by smaller apertures with the increased uncertainties expected from them. This balance established the HySICS attenuation-levels achievable via aperture ratios to $\sim 10^{-3}$.

### 4.2.3 Spectral-Scale Corrections

Since radiometric uncertainties are dependent on the product of the instrument's spectral accuracy and the derivative of the measured spectrum, knowledge of and corrections for the spectral scale (or wavelength position) are characterized and applied. Intermittent measurements using the HySICS's internal pen-ray lamp throughout the flight allow spectral calibrations based on this narrow-band source to verify spectral-scale accuracy or correct for possible wavelength-position fluctuations due to thermal or mechanical changes. Independent control of the three TECs regulating optical-bench temperature reduced thermal gradients during Flight 2 and thus reduced variations in the spectral scale. The spectral scale when at altitude shifted by only 3 nm with variations across all wavelengths being maintained to < 1 nm across the spectral range.

The spectral corrections were interpolated to the times of observations. Of particular importance are the corrections at the times of solar calibrations, as the Sun has more abrupt spectral-variations than ground scenes, and uncertainties in the spectral scale near the edge of a large spectral-variation can give a correspondingly-large radiometric-uncertainty at wavelengths near spectral lines. Since the HySICS uses a FPA that spectrally bins the incident light from the spectrometer into 3-nm regions defined by the size of





the FPA pixels, small potential spectral-shifts in the incident light coupled with large spectrally-dependent changes in signal near the sharp edges of these pixel-defined spectral bins can affect radiometric uncertainties.

Both the effects of this pixel-delineated spectral-binning and those from thermal or mechanical instrument-distortions are included in estimates of the HySICS's wavelength-position uncertainties, which are plotted as a function of wavelength in Figure 5 for solar

observations and in Figure 6 for ground-scene measurements. Because of the in-flight spectral-calibration corrections via the internal pen-ray lamp, wavelength-position uncertainties are rarely the dominant contributor to the net radiometric uncertainties, although they do increase at the shortest wavelengths, where the Sun has more spectral-absorption lines, as well as near 820 nm, where the Sun has several absorption-lines and the HySICS has low sensitivity.

### 4.2.4   Brightness Offset

The FPA has a background-level offset that varies linearly with the measured signal. This offset is detectable by observing the extreme-most ultraviolet spectral-column of the sensor, which, at 320 nm, is below the reflectivity cut-off for the instrument mirrors and therefore is effectively a dark column. All pixel-values in this column should remain nearly constant, showing mainly dark-current and fixed-pattern noise. Instead, they consistently decrease by up to 120 DN when other portions of the array are observing extremely bright signals. This background-level decrease is also observed in all dark pixels during a solar scan, including columns

neighboring the dark column as well as regions of the sensor viewing dark space up to 9.5° away from the Sun. Lab measurements of FEL and LED sources show similar effects.

This "brightness offset" of the background level, as measured on the dark column, is characterized using flight data. A matrix of background-level reduction versus sensor signal is generated from all large power-level transitions during the flight, such as when the solar disk moves out of the instrument FOV during a flat-field scan or when it comes into- or out-of-view during an irradiance

scan. The amount of background-level reduction is linear with the amount of light detected by the sensor, regardless of its spectral distribution or spatial location, with the background-level changing by $-2.3 \times 10^{-7}$ DNs per DN of signal. Because there is some dependence on the integration time of the sensor, this slope was determined for all integration times used during flight. The slope has an uncertainty of 14 % based on the standard deviation from multiple background-level characterization measurements. However, being as the brightness offset itself is a small correction, this does not directly translate into a similar-magnitude

contribution to the overall radiometric measurement uncertainties.

The resulting brightness-offset corrections, which are dependent on the total signal on the sensor as well as the integration time, are applied to each image acquired. The relative uncertainties in this correction are greatest for measurements having low signals, so predominantly affect Earth ground scenes, where they are generally the second-largest contributor to net uncertainties.

### 4.2.5   Polarization Sensitivity

Accurate radiometric measurements of scenes having unknown polarization rely on the instrument having low polarization-sensitivity. The HySICS was designed to reduce polarization sensitivity by orienting the optical plane of the 4MA perpendicularly to that of the spectrometer, such that reflection-induced diattenuation in the former is nearly offset by that in the latter. This was effective with the exception of the custom-ruled grating, the primary HySICS optical component that did not meet expected performance. Along with having low efficiency in the visible, polarization tests of this grating showed a much larger sensitivity

than anticipated, with the net instrument-diattenuation results plotted in Figure 4. Despite the orthogonal orientation of the 4MA to the Offner optics, this grating limits the instrument's desired low polarization-sensitivity particularly in the near infrared.





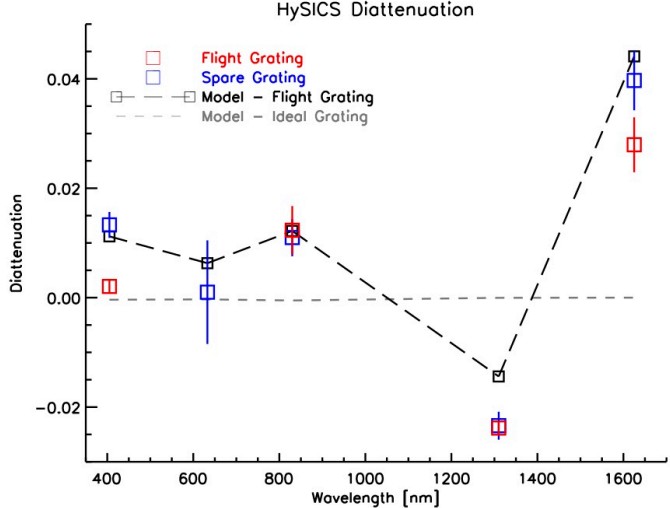

Figure 4: Diattenuation of the HySICS at the integrated-instrument level is limited by high grating-induced polarization, which is as large as 4% at wavelengths above 1000 nm. A Zemax model based on the flight-grating measurements (black, long dashes) shows the lower instrument-diattenuations expected using a less polarization-sensitive grating (grey, short dashes).

If measuring randomly-polarized scenes, this internal-instrument polarization sensitivity has no effect on radiometric accuracy, but for scenes of unknown polarization amplitude and orientation, the radiometric uncertainties can potentially be as large as the instrument's diattenuation itself in the specific – albeit highly-improbable – case of a 100%-polarized incident signal oriented along or perpendicular to the direction of the instrument's greatest polarization-sensitivity.

**4.2.6    Net Instrument-Imaging Uncertainties**

Integrated-instrument uncertainties showing the effects described above are plotted in Figure 5 and tabulated for select wavelengths in Table 3 for spatially-integrated cross-slit observations of the Sun and in Figure 6 and Table 4 for sample single-acquisition measurements of bright and dark ground-scenes. These two figures indicate the uncertainties on the measurements $S_{meas\_SSI}(\lambda)$ in Eq. (2) and $S_{meas\_obj}(\lambda)$ in Eq. (1), respectively.

With the exception of the flat-field and diffraction uncertainties, the solar-scan uncertainties benefit from multiple-image acquisitions and a dual-scan approach. Multiple, repeated measurements of the same scene particularly reduce the effects of read- and shot-noise and from the brightness-offset caused by the small vs. large apertures at the shorter wavelengths where the HySICS's response is lowest. Two back-to-back scans of the Sun, one using longer integration-times to increase signals at wavelengths shorter than 850 nm and one with integration times better matched to the higher signals at longer wavelengths, followed by spectrally-combining the scans in post-processing improves the signal in select portions of the spectrum. The improvements from multiple image-acquisitions and the dual-scan approach are possible only because the Sun can be viewed repeatedly with the same instrument look-angles, so provides a static in-flight calibration source. These techniques would also be applicable to reducing flat-field uncertainties, but were not performed on Flight 2, so the solar-calibration results shown are dominated by the flat-field uncertainties.





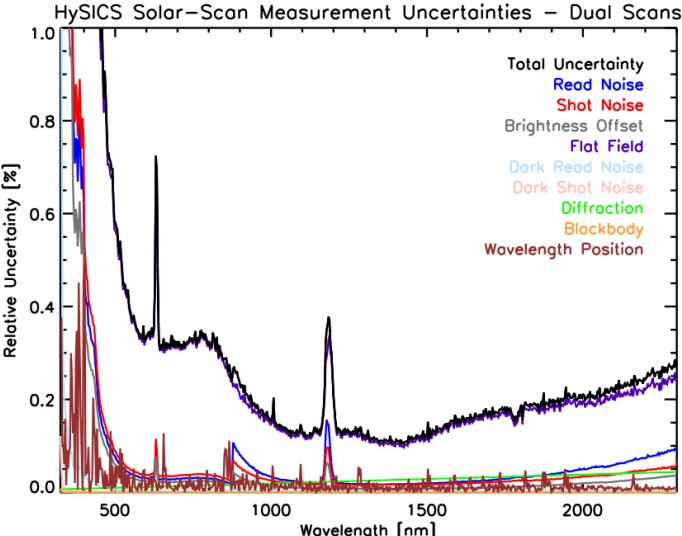

Figure 5: Contributions to net relative uncertainty (black) when observing the Sun during Flight 2 are shown as a function of wavelength. With the exception of the flat-field uncertainties, these plotted uncertainties are the result of two consecutive cross-slit solar scans acquired using specific integration-times for the short- and long-wavelength spectral-regions to reduce the uncertainties within each. Flat-field uncertainties due to low signal levels dominate across the spectrum but could be reduced with similar multi-image, dual-scan techniques applied to those calibrations.

Balloon-flight motions over the ground prevent applying these beneficial uncertainty-reduction techniques to ground scenes, so uncertainties must be based on single-image acquisitions. Despite the larger aperture and the longer integration-times for ground scenes, the lower radiances of these single images have larger relative uncertainties than those from the Sun since they do not

10   benefit from multi-image or dual-scan techniques. Typical net uncertainties from representative bright (cloud-filled) and dark (desert- and vegetation-filled) ground-scenes are plotted in Figure 6; these are the net scene-dependent uncertainties in the measurement factor $S_{meas\_obj}(\lambda)$ from Eq. (1). The dominant uncertainties are from shot- and read-noise and the brightness offsets with flat-field uncertainties dominating at shorter wavelengths. Note that while the dark scene has higher uncertainties across much of the spectral region, at the longer NIR wavelengths it has slightly greater overall signal and therefore lower uncertainties, since

15   the darker ground-scenes emit more infrared radiation than the brighter (in the visible) scenes from colder clouds; although these are still large due to the very-low reflectance-signals at these wavelengths.





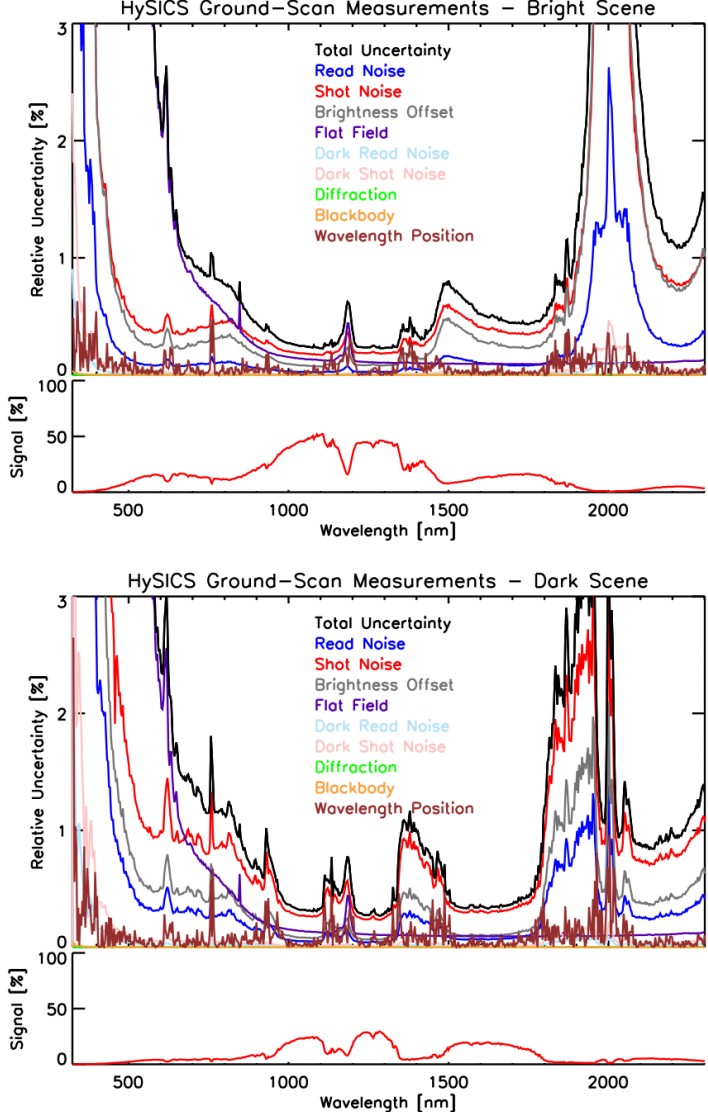

Figure 6: Contributions to net relative uncertainty (black) when observing a bright (top) and dark (bottom) Earth-scene during Flight 2 are shown as a function of wavelength. The small, inset lower plots (red) indicate the signal strength from each scene relative to full scale of the instrument's FPA. Shot noise is generally the dominant uncertainty across the majority of the spectrum for ground scenes, which do not benefit from multiple-image or dual-scan acquisition techniques.

### 4.3 Attenuation-System Uncertainties

In addition to many of the instrument-level uncertainties for various observation scenes and modes described in Sections 4.1 and 4.2, characterizing the radiometric uncertainties to which ground-scene radiances can be referenced to the spectral solar irradiance also involves quantifying the uncertainties from the three intensity-attenuation methods used to enable solar- vs. Earth-viewing. These attenuation methods, represented by the correction factor $A(\lambda)$ in Eq. (1), have additional uncertainties that are described in this section.





The total attenuations demonstrated during Flight 2 were capable of a net $10^{-7.1}$ reduction in incident radiance, much greater than the $\sim 10^{-4.7}$ needed for solar-radiance attenuations. Tables 5–7 give a numerical breakdown of the attenuation-system uncertainties for three select wavelengths across the instrument's spectral range. The dominant contributors to these uncertainties include low signal levels due to low FPA- and grating-efficiencies at certain wavelengths and high light-source variations for laboratory

calibrations in the UV and visible, which would be straightforward to improve in future calibrations. Some instrument-specific uncertainties could be reduced by decreasing the large attenuation-range demonstrated here. Forgoing the filter attenuation-system, for example, would provide a net $10^{-6.2}$ attenuation, which is still larger than required for solar viewing. Using only the other two attenuation systems has demonstrated a $\sim 2 \times$ improvement in radiometric accuracies over existing spaceflight instrumentation for an average across most of the visible and NIR spectral regions with a $\sim 6 \times$ improvement demonstrated in some regions. Further

HySICS uncertainty reductions are expected from identified improvements in lab calibrations and spectrometer grating design. Results and uncertainties from the individual attenuation-methods are detailed in the following subsections.

### 4.3.1    Aperture-Ratio Uncertainties Due to Optic-Surface-Area Illumination Differences

The baselined $10^{-3.2}$ attenuation due to aperture-area ratios was demonstrated, with measured results plotted in Figure 7. This figure also gives the corresponding uncertainties as a function of wavelength while Table 5 details a breakdown of their contributing

components. Although this aperture-ratio attenuation-method relies on geometry and so should be nearly spectrally-flat, at shorter wavelengths the method causes much more than the $10^{-3.2}$ attenuation expected from aperture-area ratios alone.

The small- and large-apertures respectively used for the Sun- and Earth-measurements illuminate different areal-portions of the HySICS optical surfaces and thus have different throughput-efficiencies that must be accounted for when transferring the solar-based radiometric scale to radiances from ground measurements. While most optical surfaces are sufficiently uniform or similarly

illuminated to not be greatly affected by these different areal-illumination effects, the spectrometer grating is the dominant cause of current HySICS spectrally-dependent efficiency-variations between the two aperture-illumination regions.

To achieve the broad spectral-range and high throughput-efficiencies required with a single-spectrometer design, varying grating-blaze-angles are needed. To stay within the HySICS's program budget, the fabricated balloon-flight grating contains four discrete regions with the blaze-angle varying similarly and monotonically across each. The small aperture used for solar measurements

illuminates a boundary between two such regions to a much greater proportional degree than the larger aperture does, so is more sensitive to symmetric alignment on this boundary. A slight misalignment on the grating boundary will preferentially favor the corresponding blaze-angle in that misalignment direction, thus making the system more sensitive to either the shortest or longest wavelengths. The relative throughput for the small and large apertures was characterized in lab measurements with the results shown in Figure 7. These effects are accounted for in HySICS's radiometric results as part of the aperture-ratio portion of the full

attenuation-system correction, $A(\lambda)$. The spectral dependence shows much lower efficiency in the visible and higher efficiency in the NIR, suggesting the small region of the grating illuminated by the solar-viewing aperture is biased toward the NIR-blaze edge of one grating-region rather than equally split between the two it straddles.

These laboratory calibrations were performed using two light sources with one peaking in the visible and the other in the NIR to span the full spectral-region. The intermediate visible-to-NIR spectral region had low intensity from both lamps. Combined with

the strong increase in attenuation and resulting lower intensities at shorter wavelengths when using the small aperture, these low light-source intensities limited the relative uncertainties in this visible-to-NIR spectral region, resulting in the large uncertainty peak shown in Figure 7. Further calibrations with a broader range of bright lamp sources, particularly near the visible-to-NIR


transition, could improve the uncertainties shown. More significantly, reducing the large spectral dependence of this aperture-ratio correction by using a continually-varying blaze-angle via a more expensive custom-made grating should reduce much of these aperture-ratio uncertainty issues in the first place, and is planned for a future HySICS instrument.

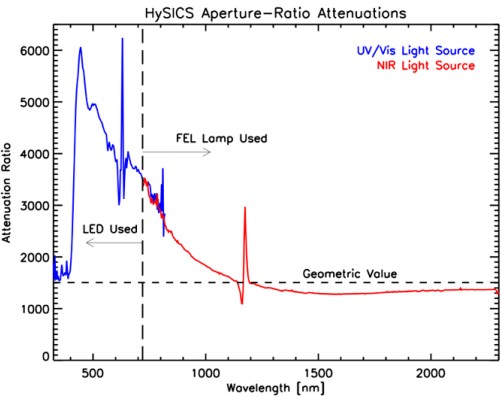
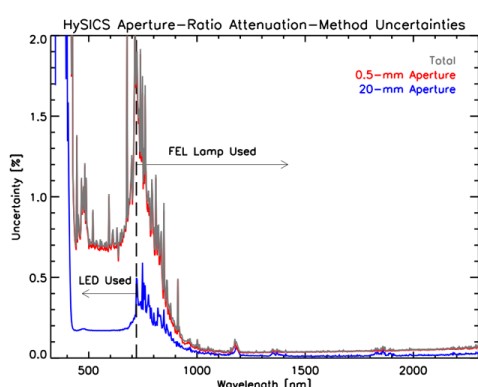

Figure 7: Different illuminations of the optical surfaces by the 0.5-mm and 20-mm apertures, mainly being affected by a boundary between grating blaze-regions, cause the spectrally-dependent attenuations that differ from the nominal geometric-ratio value due to the aperture-attenuation method, as shown in the left-hand plot. The blue and the red curves are based on different lab light-sources that provide peak power at shorter and longer wavelengths, respectively. The right-hand plot gives the uncertainties in these attenuations. The large peak in uncertainties between the two light sources
used is due to low signals from each and could be improved with additional calibration light-sources.

The aperture-ratio attenuation-technique is inherently nearly independent of wavelength. That HySICS demonstrated the technique to well less than the needed uncertainties through most of the NIR spectral region shows promise that this attenuation method would be equally applicable over the entire spectral range with a more uniformly-blazed grating and further laboratory characterizations.

**4.3.2    Integration-Time Uncertainties**

Correcting for non-linearities while varying the FPA's electronically-controlled integration times was more successful than initially anticipated, achieving a demonstrated attenuation of $> 10^{-3}$ with a 0.05 % uncertainty for generally-used exposure levels and a maximum of 0.12 % uncertainty that could accommodate extremely bright Earth-scenes. As described in Sect. 4.1.4, these corrections rely on characterizations of the electrically-controlled integration-timing signals and the FPA's resulting response to
various saturation levels.

The electronic timing signals show deviations from linearity that are $< 2 \times 10^{-7}$ for integration times from 16.8 μs to 8.62 ms, spanning a range of $10^{2.7}$ in integration times, and $< 1.6 \times 10^{-6}$ over the full range from 16.8 μs to 35.23 ms, spanning a $10^{3.3}$ range. These timing-signal deviations from linearity are relatively insignificant.

The results from the characterizations of the FPA's response described in Sect. 4.1.4, whereby the FPA's signal levels are
determined from multiple, repeated measurements of an input FEL-lamp source at different exposure times, are shown in Figure 8. Since the electronic shutter has nearly negligible non-linearity across this range, these deviations from linearity that manifest mainly at greater exposure times (i.e. greater signal levels) are due to non-linearities in the detector response and/or readout-amplifier electronics. The average of the deviations plotted in Figure 8 (lower graph) provides the applied non-linearity correction as a function of detector signal, and the standard deviations about this average give the corresponding uncertainties in the applied





non-linearity correction. The corrections are measured to be < 0.1500 % for almost all pixels over an intensity range of > $10^3$ and the reproducibility of each pixel's response is generally < 0.05 % for intensities up to 53 % (35 000 DN) of full scale, which accommodates the brightest Earth-scenes viewed during Flight 2. This attenuation method can also accommodate higher-intensity scenes, allowing up to 75 % of the FPA's full scale to be utilized while maintaining uncertainties to < 0.12 %. The corresponding

5    uncertainties are shown in Figure 9 and tabulated in Table 6.

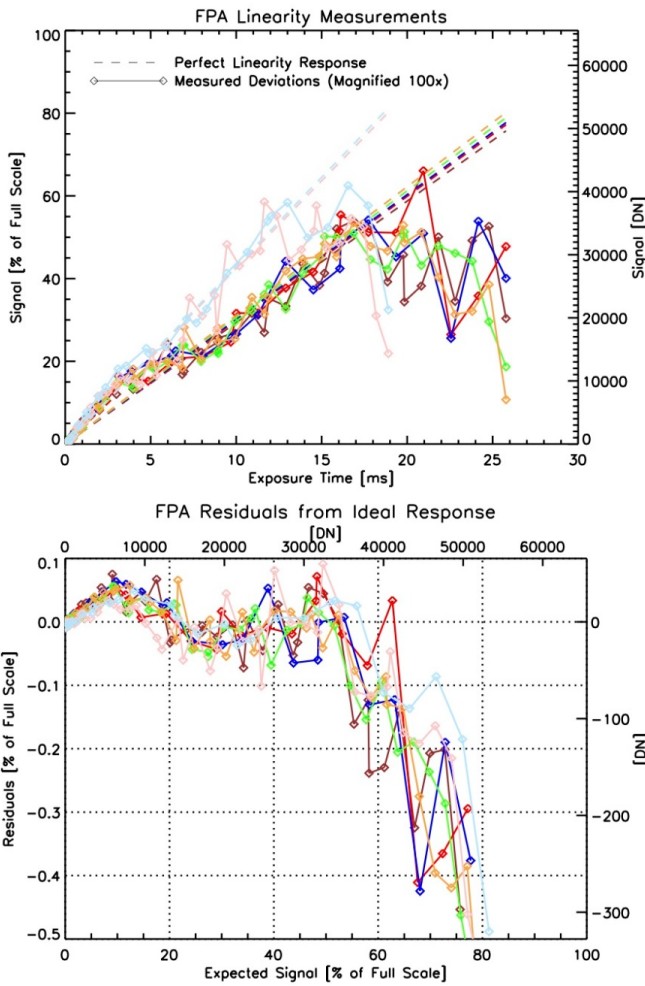

Figure 8: Detector response is plotted vs. exposure time over a range >$10^3$ for seven different linearity calibrations indicated by different colors (upper plot). The differences (solid lines) between the detector response and linear fits

10    (dashed lines), which vary between the different linearity calibrations due to intentional changes in the incident light-level, have been exaggerated 100 × for visibility. These show slight sensitivity-decreases at greater detector-signal-levels. The residuals from the fitted linearity are shown in the lower plot as a function of signal level and provide a linearity correction for measured detector-values. The repeatability of these repeated residual-measurements indicates the uncertainty to which these non-linearities can be corrected, and is shown in Figure 9. These responses are determined

15    individually for each FPA pixel.





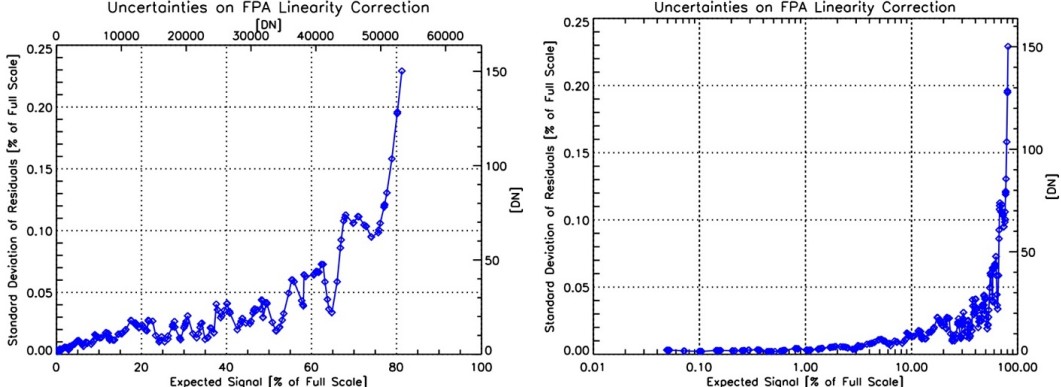

Figure 9: The standard deviations of the residuals from the average of the residuals shown in Figure 8 indicate the uncertainties in the applied linearity-correction as a function of signal level (left-hand plot). The semi-log plot of the same data demonstrates the full $> 10^3$ measured intensity-range (right-hand plot).

The greater-than-anticipated attenuation-range and the lower-than-anticipated uncertainties due to this integration-time attenuation method allow flexibility in the attenuation amounts needed by the other two attenuation methods. The integration-time attenuation capabilities provided by this flight-capable FPA eliminate the need for attenuations via filters altogether, which reduces mass, cost, complexity, and power for a future flight instrument.

### 4.3.3 Filter-Transmission Uncertainties

Spectral filters were calibrated during Flight 2 using both the Moon and the Sun. The filter attenuation-method demonstrated the desired attenuation range of $10^{-0.9}$ but with higher than the anticipated 0.05% uncertainty due to the low lunar-signal levels at the time of this flight. As explained in Sect. 4.1.6, the narrow lunar-crescent illuminated only a few pixels on the HySICS FPA, resulting in read-noise limiting filter-calibration uncertainties at wavelengths less than 900 nm (see Figure 10). Since the filter calibrations are acquired while viewing stationary sources, namely the Sun and the Moon, multiple images and different

integration-times could be used to reduce noise in low-signal portions of the spectrum, although only single integration-times were used for the calibrations during Flight 2. This operational improvement would substantially reduce the uncertainties in the visible spectral-region shown in Figure 10. Nevertheless, the filter uncertainties, particularly in the near-infrared, are already lower than those from the other two attenuation-methods (see Table 7), although this method does not provide nearly the attenuation range of either of those other two. Fortunately, the large attenuation-range provided by the integration-time attenuation-method likely makes

this entire filter-attenuation system unnecessary.





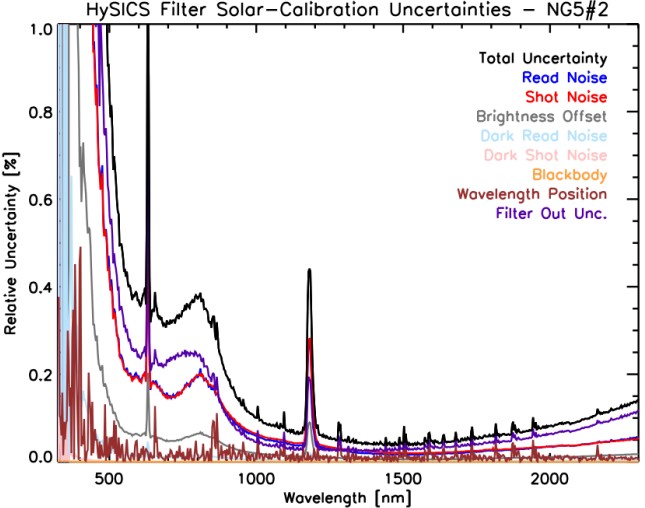

Figure 10: Calibration of NG5 #2 filter during Flight 2. Measurements with the filter out dominate the net uncertainty, as the spectrally-flatter filter-in measurements could be done at a longer integration time to achieve higher overall signal.

### 4.4 Radiometric Traceability to SI

#### 4.4.1 Scene-Reflectance Uncertainties

Sections 4.1 and 4.2 explain the intrinsic imaging-measurement uncertainties from Sun- and Earth-scenes [$S_{meas\_SSI}(\lambda)$ in Eq. (2) and $S_{meas\_obj}(\lambda)$ in Eq. (1)], while Sect. 4.3 adds uncertainties from the HySICS's attenuation-system calibrations, $A(\lambda)$, which relate the relative signals of the Earth-scenes to those of the Sun via the applied attenuation amount. This Earth-to-Sun ratio effectively gives the (unit-less) reflectance of the Earth scene. The uncertainty in the ratio includes uncertainties from the cross-slit solar-disk

scans (Figure 5), the Earth images (Figure 6), and the attenuation-systems applied (Figure 7 and Table 6 since the filter-attenuation system was not utilized for the results presented here). Being independent, these uncertainties can be root-sum-squared to give the net Earth-to-Sun ratio (reflectance) uncertainties shown in Figure 11 for both bright and dark ground-scenes.

The solar cross-calibration techniques achieved a radiometric uncertainty of nearly 0.3 % across a large spectral region long-ward of 1000 nm from a bright ground scene, demonstrating a ~6 × improvement over current spacecraft uncertainties and the capability

of the approach to achieve the desired radiometric accuracies. The net uncertainties shown in Figure 11 are dominated by spectral regions where there is very little power from the reflected-Earth radiation or very low HySICS sensitivity, such as the 1800 to 2200-nm and shorter visible spectral regions respectively, and by the increased uncertainties from the calibration of the aperture-ratio attenuations near the visible-to-NIR transition spectral-region. (The latter can be improved with further calibrations and/or an improved instrument-grating, as detailed in Sect. 4.3.1, and the former partially by improved flat-field calibrations, as explained

in Sect. 4.1.6.) At most NIR wavelengths the uncertainties are dominated by read- and shot-noise in the ground scenes. The uncertainties shown are characteristic of individual pixels; spatial or spectral binning could allow yet further reductions in uncertainties.



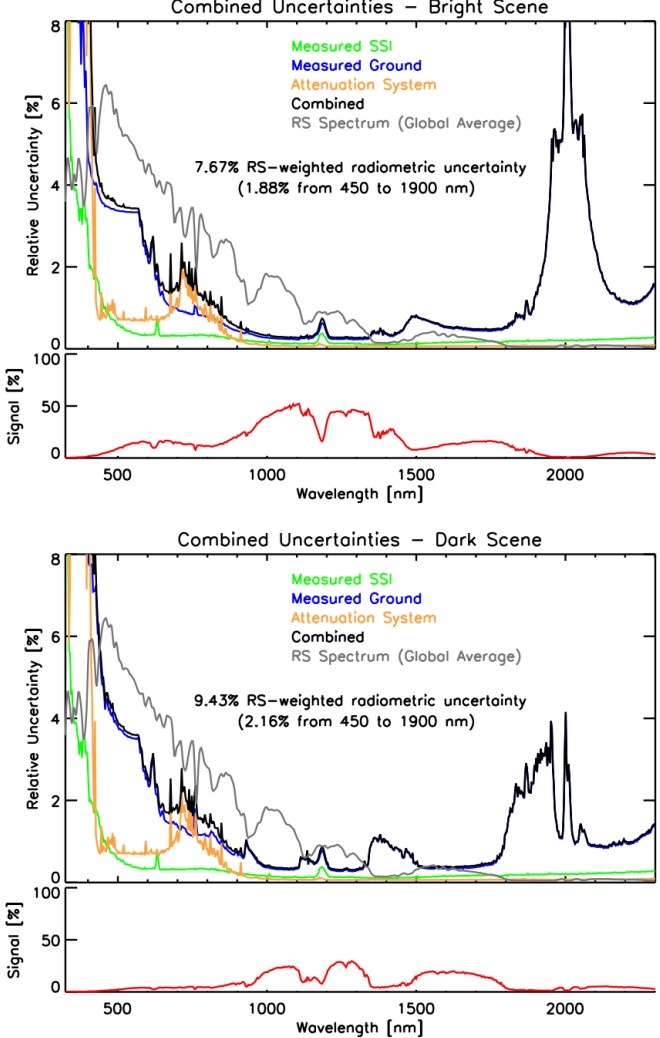

Figure 11: Contributions to net relative uncertainty (black) in the ratio of a bright (top) and dark (bottom) Earth-scene relative to the HySICS-determined SSI during Flight 2 are shown as a function of wavelength. The small, inset lower plot (red) indicates the signal strength from each scene relative to full scale of the instrument's FPA. Shown uncertainties are for individual pixels and could be reduced with spatial or spectral binning. Average uncertainties based on a weighting by globally-averaged reflected-solar radiances are given over full (350 to 2300 nm) and partial (450 to 1900 nm) wavelength ranges. Demonstrating a minimum uncertainty of ~0.3 % at wavelengths longer than 1000 nm indicates that the solar cross-calibration method used by the HySICS has promise of meeting the desired radiometric accuracies.

A globally-averaged, all-sky estimate of Earth-reflected irradiance over the eight-year period from 2003 to 2010 based on results from observation-system-simulation-experiments generated using SCIAMACHY data [15] is plotted in Figure 11 to indicate the realistic reflected-solar (RS) spectrum observed by a spaceflight hyperspectral imager. Weighting the HySICS's net radiometric-uncertainties by this estimated spectral-irradiance gives the resulting spectrally-averaged radiometric uncertainties in the figure. These are higher than ultimately desired, which is largely caused by low instrument efficiencies and high flat-field uncertainties in the visible as well as increased aperture-ratio attenuation uncertainties near the visible-to-NIR transition. Improving these via the methods described in Sect. 4.3.1 and extending the multiple-image acquisition and a dual-scan approach to flat-field calibrations




should reduce the weighted, Earth-reflected HySICS uncertainties for a future instrument by another ~4 × improvement over the values demonstrated here.

### 4.4.2    Conversion to Physical Units with SI-Traceability

The high-quality data from Flight 2 with all instrument-level and attenuation-method corrections applied and with a final
calibration-factor, $C(\lambda)$ in Eq. (1), to provide an SI-traceable radiometric-scale enables the creation of three-dimensional spatial/spectral data cubes from ground scans that represent the end product of the HySICS solar cross-calibration technique. With sufficiently low uncertainties in the three factors in Eq. (1), this results in hyperspectral ground-images with lower SI-traceable radiometric-accuracy uncertainties than existing flight instruments provide. The details of acquiring this final calibration factor are described in this sub-section.

Multiple images as the solar disk is scanned in the cross-slit direction are spatially integrated to give a net spectral solar irradiance with corrections to account for the spectrometer's NIST-calibrated slit-width as well as image overlap during the cross-slit scan. This irradiance is corrected for the diffraction and scatter described in §4.2.2 as well as other instrument effects described above. At this stage, the spectral "irradiance" is in units of instrument data-numbers (DNs) and has no traceability to normal physical units. Figure 12 shows the SSI determined from HySICS using a cross-slit scan of the solar disk during Flight 2.

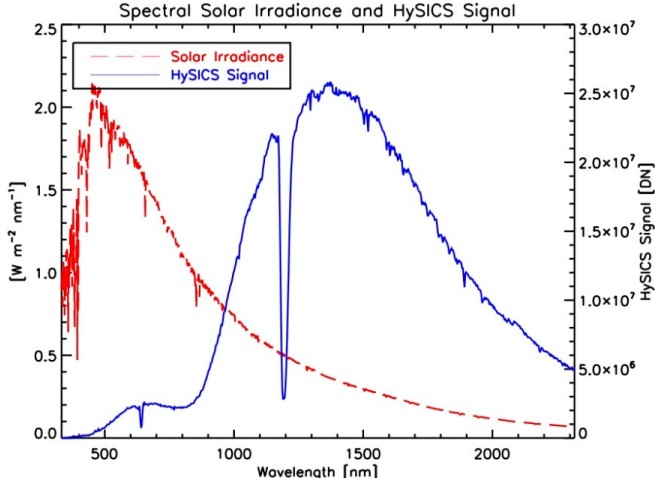

Figure 12: The HySICS signal from spatially-integrated cross-slit solar scans with all applied instrument-level corrections gives an instrument-level spectral solar "irradiance" (blue). The values are given in instrument DNs and have no traceability to SI at this stage. That traceability and physical units are provided by scaling to the NRLSSI2 model for the day of the flight. These values, plotted in red, are adjusted to the Sun-instrument distance at the time of Flight 2 to correctly
indicate the actual SSI that should be measured at the HySICS's location. These, or direct solar-measurements from space-borne instruments having high absolute-accuracies, enable the SI-traceable cross-calibration of the HySICS-measured SSI. (Note that the DN values exceed the 16-bit maximum values from individual pixels because the plotted HySICS SSI signal is the spatially-integrated sum of the entire solar disk.)

By knowing what the actual SSI is, the HySICS instrument DNs can be converted to useful physical units and the overall instrument
sensitivity can be determined. SSI values from Dr. Judith Lean's NRLSSI2 model were applied to the HySICS Flight 2 data since they were available prior to measurements from any on-orbit instrument. These values, which account for the solar activity state on that day, were adjusted from their as-provided 1-AU distance to the actual Earth-Sun distance on the date of the flight. They are plotted in Figure 12 and provide the transfer to realistic physical units.





The ratio of this model-based "actual" SSI to the HySICS's measured "irradiance" in Figure 12 gives the instrument sensitivity via a conversion from DNs to physical irradiance units via correction factor $C(\lambda)$, as shown in Eq. (2). This conversion factor is plotted in Figure 13. The low instrument-sensitivity in the visible is mainly the result of low efficiencies of the FPA and the grating at these wavelengths.

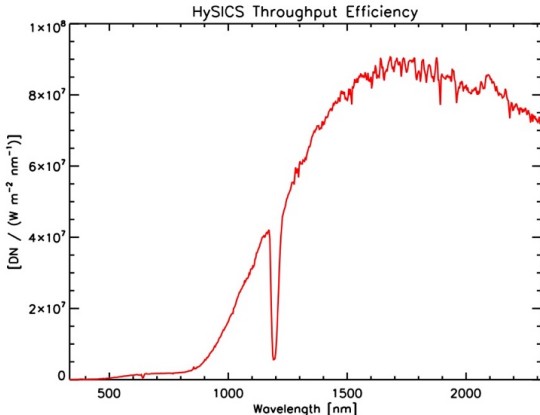

Figure 13: Dividing the HySICS-measured spectral solar irradiance by the modeled SSI in Figure 12 gives the effective end-to-end sensitivity of the instrument with SI-traceability via correction factor $C(\lambda)$. This cross-calibration can then be applied to Earth ground-scene observations after correcting for the HySICS's attenuations applied via factor $A(\lambda)$ to provide SI-traceable shortwave-reflected Earth-radiances via Eq. (1).

10  **5    HySICS Results: Radiometrically-Calibrated Data-Cubes**

Applying the conversions $A(\lambda)$ (correcting for the differences in attenuations between the solar measurements and the ground measurements) and $C(\lambda)$ (converting HySICS DNs into radiance units) to the HySICS-measured ground-scenes $S_{meas\_obj}(\lambda)$ in Eq. (1) gives the resulting radiometrically-calibrated ground-scene shown in Figure 14 for a single wavelength. This approach is applied at all HySICS wavelengths and provides a full three-dimensional data-cube having SI-traceable radiometric accuracy, such

15  as the example of a ground scene shown in Figure 15 and the lunar scan in Figure 16. Such radiometrically-calibrated data-cubes are the desired final products of the HySICS, and this improved-accuracy technique based on solar cross-calibrations has now been successfully demonstrated via Flight 2.



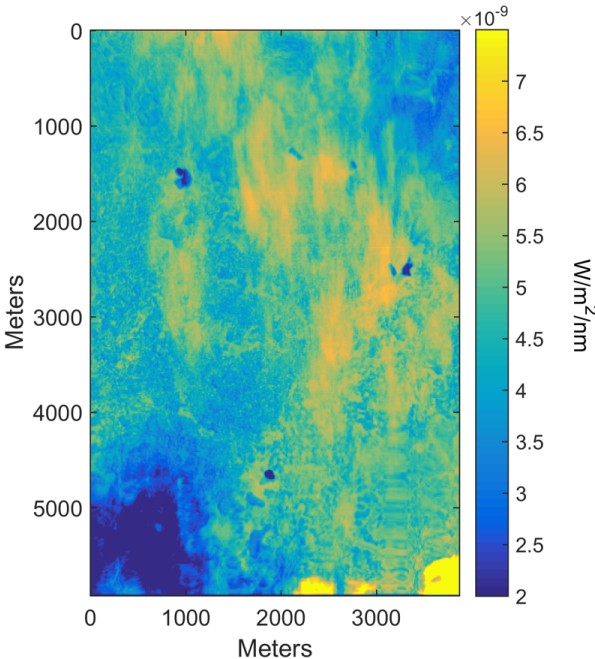

Figure 14: This two-dimensional spatial ground-scene at 1233 nm from Flight 2 is radiometrically calibrated using the conversion values from Figure 13.

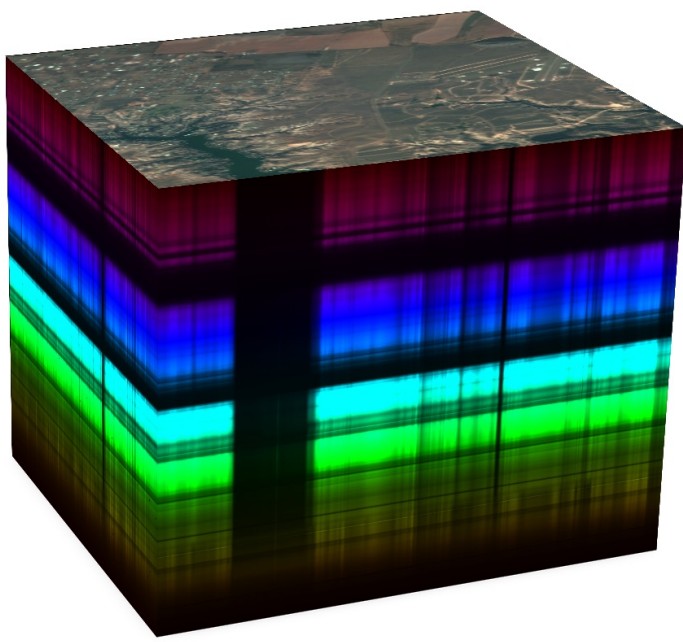

Figure 15: This representation of a three-dimensional data-cube is from a ground-scene scan over mixed desert and water to show the generic HySICS data products of spatial/spectral imagery.



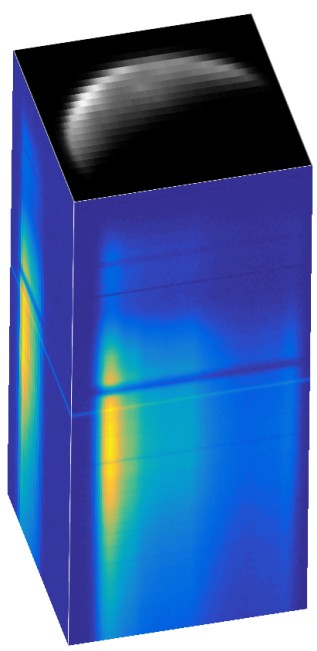

Figure 16: Data cubes from HySICS's in-flight lunar scans provide spectral-radiance as well as spatially-integrated irradiance measurements of the Moon (as viewed from the Earth) with improved radiometric accuracy than has yet been obtained from Earth-based measurements.

## 5   6   Conclusions and Spaceflight Potential

Built under a NASA ESTO Instrument Incubator Program, the HySICS uses direct radiance-measurements of the Sun to cross-calibrate hyperspectral images of other scenes, such as of the ground, Earth's atmosphere, or the Moon, with improved radiometric accuracies over similar instruments relying on indirect or diffused solar-observations, on-orbit light-sources, pre-launch calibrations, or measurements of vicarious ground-sites. This measurement technique, utilizing three precisely-characterized

intensity-attenuation methods, enables direct in-flight calibrations relative to the spectral solar irradiance, which is a more stable and better-known reference than other space-based light sources, and allows SSI measurements to benchmark Earth ground-scenes with radiometric accuracy and long-term precision greatly exceeding the capabilities of current space-based ground-imaging instruments. The demonstrated improvements from the HySICS were accomplished using the instrument's optical-design covering the entire reflected-solar spectrum and based on a single flight-capable FPA, demonstrating reduced mass, volume, power, and

cost for air- or space-flight instrumentation compared to multi-focal-plane designs intended to cover this broad spectral region.

The HySICS's solar cross-calibration methods have been applied to provide radiometrically-accurate, SI-traceable spatial/spectral data-cubes of ground scenes and the Moon from two high-altitude balloon flights. Using all three of its intensity-attenuation systems, the HySICS achieved net radiometric intensity-reductions of $10^{-7.1}$, exceeding the $10^{-4.7}$ attenuation capability required to enable direct-view solar cross-calibrations. Operating over the more limited – yet still easily sufficient – $10^{-6.2}$ attenuation-range

provided by the HySICS's aperture-ratio and integration-time attenuation methods eliminates the need for the complexities of the HySICS's filter-based attenuation-system for a spaceflight instrument. Although demonstrating all three intensity-attenuation methods, the second of two high-altitude balloon flights of the HySICS demonstrated a ~2 × improvement in radiometric accuracies



over most existing spaceflight spectral-imagers across a reflectance-weighted average of the visible and NIR and select spectral regions achieving a ~6 × improvement using only the aperture-ratio and integration-time attenuation methods. An additional radiometric-accuracy improvement of ~4 × could be expected for the existing instrument via better aperture-ratio lab-based calibrations and multiple in-flight flat-field calibrations. These two effects were identified as being the dominant contributors to

radiometric uncertainties of ground scenes across most of the observed spectral region, and both would be relatively straightforward to improve in future measurements and calibrations. Yet further improvements are expected via identified spectrometer-grating design changes.

Radiometric uncertainties based on the HySICS's solar cross-calibration approach were characterized as a function of wavelength for the balloon-flight data. The largest uncertainties were identified as being due to: FPA and grating efficiencies in the visible,

which cause dominant flat-field uncertainties as well as high shot- and read-noise; limited light sources used in laboratory calibrations of attenuations due to the aperture-ratio method; and the low lunar-signals during the times of the balloon flights. The quantified HySICS uncertainties were not limited by any intrinsic aspect of the solar cross-calibration approach, as demonstrated by the minimum pixel-level uncertainty of ~0.3 % from a bright ground scene. This HySICS solar cross-calibration approach thus shows promise to ultimately achieve the ~10 × radiometric-accuracy improvements desired for future climate studies with the

instrument design improvements that have been identified and will be implemented in future flight instruments.

Versions of the HySICS have been designed for accommodation on free-flyer spacecraft as well as the International Space Station, with either platform offering future instrument-opportunities and the acquisition of scientifically-valuable data. Studies are currently underway to manifest the HySICS on the CLARREO Pathfinder mission to improve spaceflight technology-readiness, demonstrate the ability to achieve eventual CLARREO-mission climate-benchmark measurement requirements, and provide inter-

calibrations of other on-orbit sensors. The CLARREO Pathfinder / HySICS is planned for launch to the International Space Station in 2020.

## 7     Data Availability

This paper discusses the corrections and uncertainties that have been characterized from a variety of lab- and flight-based data to ultimately provide improved radiometric accuracies for future spaceflight hyperspectral imagery. With that emphasis on acquiring

high-quality calibration data, only a few such final representative images were acquired during the limited flight time. The corrections described in this paper are being applied to these final HySICS balloon-flight data products of representative ground- and lunar-scans, such as shown in Figures 14 and 16, and corresponding uncertainties to those data-cubes are being produced. Those data, which include a few ground-scan scenes of both desert and clouds, an Earth-limb scan, and a lunar scan, will be available via request to the authors when completed, but are auxiliary to and not the primary focus of this paper.

The actual calibration data themselves are diverse, with flight data coming from the instrument's FPA and internal sensors, the WASP pointing system, and numerous other balloon-flight sensors, and with laboratory data coming from several light-source monitors, temperature sensors, and hand-written lab notebooks recording specific conditions during tests. No attempt has been made to consolidate these data for simple online distribution. Instead, this paper details the processes utilized and the results achieved from a combination of those many sources over the few years of laboratory testing and characterizations subsequent to

and after the balloon flight.



**Author Contributions**

All manuscript authors actively contributed to the HySICS or WASP calibrations, operations, and/or data analysis during and following Flight 2. Dr. G. Kopp was the principal investigator of the HySICS and did the majority of the writing as the primary author of the manuscript. Second author P. Smith completed the data analyses presented here and contributed directly to the writing,
designed the flight operations software, and performed ground calibrations. G. Drake was the project manager and coordinated efforts between the HySICS team and the Columbia Scientific Balloon Facility for Flight 2. J. Espejo was the optical designer and verified instrument performance prior to launch and during flight. C. Belting, Z. Castleman, and K. Heuerman designed the electrical- and mechanical-interfaces and control systems for the flight instrument, verified operation prior to launch, and performed ground calibrations. J. Lanzi and D. Stuchlik led the NASA/WFF WASP team that enabled the pointing accuracies
needed for tracking the Sun and Moon and performed WASP operations during flight. The listed authors are most directly responsible for the results presented here, although many others at LASP and WFF contributed to the success of the HySICS balloon flights.

**Acknowledgements**

We greatly appreciate the support of the NASA/WFF WASP team led by D. Stuchlik and J. Lanzi that enabled the HySICS pointing
capabilities needed to demonstrated the instrument's solar cross-calibration approach.

This effort was funded by NASA's Earth Science Technology Office's Instrument Incubator Project under contract NNG04HZ05C as IIP-10-0019, and their enabling support is also greatly appreciated.

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




**Tables**

### Table 1: HySICS Performance Specifications

| Parameter | Value |
|---|---|
| Effective Focal Length (EFL) | 82.2 mm |
| Field of View (FOV) | 10° |
| Instantaneous FOV (IFOV) | 0.02° |
| Point Spread Function (PSF) | 90% energy in 30 μm pixel |
| Average Slit Width | 28.297 μm |
| Offner Magnification | 1:1.006 (object:image) |
| Spectral Range | 350-2300 nm |
| Spectral Resolution | 6 nm, constant, Nyquist-sampled |
| Aperture Diameters | 20, 10, and 0.5 mm |
| Nominal Frame Rate | 14 Hz |

### Table 2: Flat-Fielding Uncertainties

| Parameter | Measurement Uncertainty (%) | | | Measurement Uncertainty (%) | | |
|---|---|---|---|---|---|---|
| | 550 nm | 1000 nm | 2000 nm | 550 nm | 1000 nm | 2000 nm |
| | Solar Flat-Field | | | Lunar Flat-Field | | |
| Peak Variation Uncertainty | 0.41 | 0.15 | 0.18 | 4 | 1.1 | 1 |
| Pointing Accuracy | 0.065 | 0.059 | 0.063 | 0.32 | 0.28 | 0.28 |
| Blackbody Radiation Correction | 0 | 0 | 0 | 0 | 0 | 0 |
| Background Level Correction | 0.053 | 0.007 | 0.012 | 0.007 | 0.001 | 0.001 |
| **Total** | **0.418** | **0.161** | **0.191** | **4.013** | **1.135** | **1.038** |

### Table 3: Solar-Irradiance Scan Uncertainties

| Parameter | Measurement Uncertainty (%) | | |
|---|---|---|---|
| | 550 nm | 1000 nm | 2000 nm |
| **Image-Dependent Uncertainties (Solar Scans)** | | | |
| Read Noise | 0.044 | 0.034 | 0.049 |
| Shot Noise | 0.054 | 0.026 | 0.033 |
| Flat-Field Correction | 0.41 | 0.15 | 0.19 |
| Hot Pixel | 0.0003 | 0.002 | 0.0005 |
| Wavelength Bin Location | 0.027 | 0.015 | 0.009 |
| Blackbody Radiation Correction | 0.0001 | 0.0001 | 0.0001 |
| Background Level Correction | 0.035 | 0.014 | 0.019 |
| Dark Image Read Noise | 0.003 | 0.002 | 0.004 |
| Dark Image Shot Noise | 0.0001 | 0.0001 | 0.0001 |
| Diffraction (0.5 mm Aperture) | 0.01062 | 0.018 | 0.0378 |
| Pointing Accuracy | 0.011 | 0.011 | 0.011 |
| **Total** | **0.419** | **0.159** | **0.204** |




**Table 4: Ground-Scan Uncertainties**

| Parameter | Measurement Uncertainty (%) | | | Measurement Uncertainty (%) | | |
|---|---|---|---|---|---|---|
| | 550 nm | 1000 nm | 2000 nm | 550 nm | 1000 nm | 2000 nm |
| **Image-Dependent Uncertainties (Ground Scans)** | | Bright Pixel | | | Dark Pixel | |
| Shot Noise | 0.41 | 0.21 | 3.9 | 1.11 | 0.34 | 1.46 |
| Read Noise | 0.097 | 0.032 | 1.5 | 0.38 | 0.075 | 0.57 |
| Flat-Field Correction | 3.1 | 0.155 | 0.098 | 3.1 | 0.155 | 0.098 |
| Diffraction (20 mm Aperture) | 0.000144 | 0.00036 | 0.00054 | 0.000144 | 0.00036 | 0.00054 |
| Wavelength Bin Location | 0.041 | 0.029 | 0.12 | 0.043 | 0.066 | 0.28 |
| Background Level Correction | 0.29 | 0.098 | 4.4 | 0.58 | 0.11 | 0.86 |
| Blackbody Radiation Correction | 0.0002 | 0.0001 | 0.004 | 0.0004 | 0.0001 | 0.0006 |
| Dark Image Read Noise | 0.007 | 0.002 | 0.1 | 0.027 | 0.005 | 0.04 |
| Dark Image Shot Noise | 0.019 | 0.007 | 0.266 | 0.06 | 0.014 | 0.089 |
| **Total** | **3.142** | **0.282** | **6.077** | **3.366** | **0.402** | **1.815** |

**Table 5: Aperture Attenuation-Method Uncertainties**

| Uncertainty Parameter | Measurement Uncertainty (%) | | |
|---|---|---|---|
| | 550 nm | 1000 nm | 2000 nm |
| Read Noise (0.5-mm Aperture) | 0.054 | 0.005 | 0.0018 |
| Read Noise (20-mm Aperture) | 0.013 | 0.003 | 0.001 |
| Shot Noise (0.5-mm Aperture) | 0.12 | 0.012 | 0.004 |
| Shot Noise (20-mm Aperture) | 0.0095 | 0.003 | 0.002 |
| Diffraction (0.5-mm Aperture) | 0.01062 | 0.018 | 0.0378 |
| Diffraction (20-mm Aperture) | 0.000162 | 0.000288 | 0.000558 |
| Dark Image Read Noise (0.5-mm Aperture) | 0.0038 | 0.0004 | 0.0001 |
| Dark Image Read Noise (20-mm Aperture) | 0.0009 | 0.0002 | 0 |
| Dark Image Shot Noise (0.5-mm Aperture) | 0.0003 | 0.0001 | 0.0001 |
| Dark Image Shot Noise (20-mm Aperture) | 0.0001 | 0 | 0 |
| Background Level Correction (0.5-mm Aperture) | 0.0006 | 0.0015 | 0.0003 |
| Background Level Correction (20-mm Aperture) | 0.0007 | 0.0038 | 0.0016 |
| Light Source Variation (0.5-mm Aperture) | 0.68 | 0.022 | 0.007 |
| Light Source Variation (20-mm Aperture) | 0.17 | 0.012 | 0.004 |
| Measurement Variation (0.5-mm Aperture) | 0 | 0.058 | 0.031 |
| Measurement Variation (20-mm Aperture) | 0 | 0.037 | 0.016 |
| Dark Image Offset | 0 | 0.88 | 0.88 |
| Short Exposure Uncertainty | 0.78 | 0.78 | 0.78 |
| **Total** | **0.713** | **0.077** | **0.052** |



**Table 6: Integration-Time Attenuation-Method Uncertainties**

| Uncertainty Parameter | Bright Scene (53% FS) [%] | Max. Int. (75% FS) [%] |
|---|---|---|
| Electronic Linearity | 0.00016 | 0.00016 |
| Gain Non-linearity | 0.050 | 0.120 |
| **Total** | **0.050** | **0.120** |

**Table 7: Filter-Calibration Attenuation-Method Uncertainties**

| Parameter | Measurement Uncertainty (%) | | | Measurement Uncertainty (%) | | | Measurement Uncertainty (%) | | |
|---|---|---|---|---|---|---|---|---|---|
| | 550 nm | 1000 nm | 2000 nm | 550 nm | 1000 nm | 2000 nm | 550 nm | 1000 nm | 2000 nm |
| **Filter (Solar Calibration)** | | NG4#2 | | | NG5#2 | | | BG25 | |
| Shot Noise | 0.94 | 0.1 | 0.047 | 0.25 | 0.065 | 0.035 | NA | 0.027 | 0.018 |
| Read Noise | 0.97 | 0.097 | 0.048 | 0.25 | 0.06 | 0.035 | NA | 0.015 | 0.011 |
| Wavelength Bin Location | 0.027 | 0.015 | 0.009 | 0.027 | 0.015 | 0.009 | 0.027 | 0.015 | 0.009 |
| Filter-out Uncertainty | 0.35 | 0.049 | 0.064 | 0.35 | 0.049 | 0.064 | 0.35 | 0.049 | 0.064 |
| Background Level Correction | 0.23 | 0.023 | 0.011 | 0.082 | 0.02 | 0.011 | NA | 0.005 | 0.003 |
| Blackbody Radiation Correction | 0.0001 | 0 | 0 | 0 | 0 | 0 | NA | 0 | 0 |
| Dark Image Read Noise | 0.069 | 0.007 | 0.003 | 0.018 | 0.004 | 0.002 | NA | 0.001 | 0.0008 |
| Dark Image Shot Noise | 0.003 | 0.0003 | 0.0001 | 0.0007 | 0.0002 | 0 | NA | 0 | 0 |
| **Total** | **1.416** | **0.150** | **0.094** | **0.505** | **0.104** | **0.082** | **0.351** | **0.060** | **0.068** |

