# Peer review of "Radiometric flight results from the HyperSpectral Imager for Climate Science (HySICS)"

_Geoscientific Instrumentation, Methods and Data Systems, 2016_

## Referee Comment (RC1) · Anonymous Referee #1 · 4 Jan 2017

i find the paper very well written, with a high scientific level.

I just have some very small comments: * §2.1: i recommend for a better understanding of the optical system to introduce a figure describing the instrumentation * figure 1: change burring to blurring in the legend * §4.2.3.: a way to have a better estimation of the spectral scale correction is also to use the 02 atmosphere absorption band which very narrow, this will reduce the uncertainty of this correction as two different methods can be used with independent errors. * figure 4: why the x axis does not cover the 2-2.5 $\mu$m spectral range? * spectral shift: i understand the way to achieve a spectral calibration (positioning) is done using a lamp which marginally covers the SWIR band. What is happening in the domain? * to reduce the flat field uncertainty it is said that a multi image processing could be a way. This is generally what is done why not doing it directly, before processing?

---

## Author Comment (AC1) · 23 Jan 2017

23 Jan. 2017

Thank you very much for your helpful comments, and I very much appreciate your careful reading of the article!

Addressing each of your points:

1) We will add a schematic of the instrument optical layout. That's a good idea. I've currently left a placeholder for the figure for when we get it completed.

2) Opps – thanks! It's done.

3) I've added a paragraph to the end of 4.3.2 to mention this. It offers a verification method under actual observation conditions, which could be helpful for an on-orbit

instrument.

4) Figure 4 shows the actual diattenuation measurements and the resulting model based on them. The measurements themselves do not extend above 1600 nm, therefore I only showed the spectral scale going that far (rather than compress the results by adding blank space to the plot at longer wavelengths).

5) Containing both Hg & Ar, this pen-ray lamp does provide spectral lines through the SWIR too. In the paper's comments section, we've attached a supplemental plot of the measured spectrum from this lamp, showing several lines in the NIR. The signals are higher than you might expect, being weighted by the HySICS-instrument efficiencies, which are also higher in the NIR. (This plot is more just to satisfy your curiosity, but isn't something I think worth adding to the paper itself.)

6) The multiple image-acquisitions and the dual-scan approach we mention needs to be done at the time of the observations; it's not merely a ground-based post-processing step. The dual-scan approach relies on doing two consecutive scans of the Sun (or Moon) using different FPA integration times for each, such that a short integration-time scan contains unsaturated data (for obtaining the signals from spectral regions having high instrument-efficiency and signal) and the second scan having longer integration times to increase the signals from low-efficiency spectral regions. The flat fields can then be obtained in the separate spectral regions using the dual-scan images appropriate for each region. This approach could lower flat-field uncertainties had we performed it during the flight. (In the current analysis, we have already benefitted from reducing uncertainties via any multiple, redundant images acquired, but we can't retroactively acquire data with the long and the short integration-times.)

Thanks again for the good comments!

Greg

Please also note the supplement to this comment:

http://www.geosci-instrum-method-data-syst-discuss.net/gi-2016-37/gi-2016-37-AC1-supplement.pdf

Interactive
comment

[Figure]

**Supplement:**

---

## Referee Comment (RC2) · Anonymous Referee #2 · 24 Jan 2017

**General Comments:**

This paper describes results from early flight tests of the HySICS sensor. The paper is well written, well organized, and is not over-sold. Additionally, the paper points out shortcomings and possible improvements to the instrument. The data are well presented and support the discussion. This reviewer has few criticisms. Ideally, it would have been nice to have had better/more data to work with to better explore the uncertainties associated with flat-fielding, but of course one must work with the data one has. There were a couple minor typographical errors and at least some sections that were confusing (at least to me). These are described below. Although this paper addresses a sensor that is still "a work in progress," and thus may be considered of limited interest, this paper is important and warrants publication because the HySICS sensor promises order-of-magnitude reduction in critical spectral irradiance uncertainties.

Comment on the following: Like probably all researchers, having suffered through poor comments from reviewers, I strongly suggest the authors recognize the fallibility of this reviewer and do not exert undue effort addressing my comments that are trivial or off-the-mark. Hopefully some of the comments on a very nice paper are useful.

Additional comment: It appears there are at least two versions of the manuscript, and thus my comments may be incorrectly referenced for the most current version.

**Individual issues/questions:**

1. Page 3, line 16. Is there a concern about the durability of carbon nanotubes? In particular, is there a concern these will break off in launch and contaminate the instrument? (Addressing this question is probably not needed in this paper, just a thought.)
2. Page 3, line 20 & 21. I was confused by the statement "smoothly-varying blaze-angle." I gather there was actually more of a "step-function" between regions of the blaze angle on the grating. Note: The problem with the current grating is addressed on p. 21, line 2, in the phrase "… by using a continually-varying blaze-angle …" Is there a distinction between "smoothly-varying" and "continually-varying"?
3. Broader question: With the aperture approach, is there a long-term degradation concern about portions of the optics interacting with the solar signal and other portions of the optics not interacting with the solar signal? (Answering this may beyond the scope of this paper.)
4. Comment: Very nice discussion of the calibration factors on p. 8.
5. Page 13, lines 11 & 12: It appears diffraction results are shown in the bottom of Fig. 2 as well as in Fig. 3. I thought a bit more description of the diffraction pattern on the bottom of Fig. 2 was warranted (even though I think I understand it).
6. Question: Are "fixed pattern noise" and "flat-fielding uncertainties" basically referring to the same thing? Most of the discussion seems to focus on "flat field

uncertainties," but the phrase "fixed pattern noise" is used a couple times (at least) Sec. 4.4.4, line13, & p. 10, line 3.

7. Page 25, sentence beginning at line 10: I find this sentence confusing – seems to say Earth-reflected irradiance is plotted in Fig.11. Should this be the uncertainty in earth-reflected irradiance?

**Technical corrections:**

1. I did not see where the "SSI" acronym was defined. As this is clearly important to the discussion, it should be defined in spite of being well known.
2. Of less importance, the NPOESS acronym is also not defined.
3. Page 4, line 20. Is the word "apertures" missing between "installed" and "have"?
4. Page 13, line 4 in Figure caption: Typo "burring" should be "blurring"?

---

## Author Comment (AC2) · 24 Jan 2017

[revised manuscript text omitted]

Figure 1: Optical layout of the HySICS shows the 4MA telescope followed by a grating-based Offner spectrometer that images onto a full-spectral-range HgCdTe focal-plane-array with a three-region order-sorting filter on the back surface of its vacuum entrance-window. The Offner and 4MA have nearly-orthogonal optical-axis planes to reduce polarization sensitivity. The main picture shows a top view of the entire optical path while a side view of the 4MA itself is shown in the upper right inset. The physical entrance-aperture is positioned at the system's aperture stop. The spectrometer entrance slit is shown in its correct (albeit unconventional) orientation.

[revised manuscript text omitted]

For a spaceflight instrument regularly acquiring Earth observations, the spectral scale determined by the pen-ray calibrations could be validated by select Earth-atmospheric spectral lines under certain viewing conditions to help distinguish them from surrounding spatial or spectral features, such as by observing these lines from uniform bright background clouds or dark oceans or viewing them near the Earth limb by off-pointing from nadir. Oxygen molecules provide some such possible spectral lines, with one
5    HySICS balloon flight even showing an $O_2$ line in emission against darker space in an Earth-limb scene. Although spectral-scale corrections are a small source of uncertainty and predominantly affect solar calibrations rather than Earth observations, such Earth-atmospheric spectral-line observations could help verify the instrument's in-flight spectral scale during normal observations.

[revised manuscript text omitted]
** | **1.416** | **0.150** | **0.094** | **0.505** | **0.104** | **0.082** | **0.351** | **0.060** | **0.068** |

---

## Author Comment (AC3) · 3 Feb 2017

[The supplement with this comment has the latest version of the paper and addresses the suggestions from both reviewers.]

Dear Reviewer #2,

Thank you very much for these helpful comments, and also for the compliments on the paper itself presented in the friendly, unassuming nature of your review! Such personal pleasantries have made this paper's peer-review process very enjoyable.

Addressing each of your issues and questions (hopefully with the same unassuming demeanor):

1) Vibration of CNTs would be a potential concern for a rocket-launched spacecraft

instrument, although tests so far (done by others, not us) indicate CNTs are very robust and do not particulate. I have another program that will be testing CNTs for direct solar-irradiance applications, where the CNTs will be subjected to those applications' more extreme vibe, thermal-vacuum, and UV-radiation tests, but that's not done yet. For the low-exposure use on this relatively non-critical coating on the backside of the slit inside the HySICS's spectrometer in the much more benign balloon-launch environment described in this paper, a vibe concern is probably more detail than should be added to the sentence, which is the only mention of CNTs in the paper.

2) I've elaborated on the description of the grating, explaining it as a "saw-tooth" pattern of four "teeth," each being a region of smoothly- and monotonically-varying blaze angles. The proposed improved grating would avoid the discontinuity in blaze angle at the edge of each tooth by varying smoothly but non-monotonically several times across the entire grating, with each optimized at different wavelengths spanning the spectrum.

You're right, "smoothly-" and "continually-" varying are the same the way I used them. I've replaced the one use of "continually" with "smoothly" to avoid any potential confusion.

3) With the focus of this paper being on the results of our short-duration balloon flight #2, I agree, long-term degradation concerns such as these are beyond the scope of this paper, but would be quite appropriate for a description of the design for a spaceflight instrument.

Still, a short (and only partial) answer to your question is that this is the inverse of the issue we have with our solar-viewing instruments, which look at the Sun nearly all the time with a different aperture than they use for calibrations, so they provide lots of opportunity to have solar UV bake contamination onto the post-aperture optics with different patterns for the solar- and calibration-apertures. On HySICS, we look at the ground nearly all the time with only intermittent solar observations. Those solar observations with the 500-micron aperture don't let in much light to start with, and that

light is spread by the 0.5-degree spatial extent of the Sun before reaching the first optic. The combination of the small aperture and the infrequent solar exposure reduces some concerns about the long-term degradation.

Nevertheless, I do have plans to track any such degradation for a spaceflight instrument, but that more complete answer takes much more time to explain...

4) Thank you!!!

5) Good idea. I've added more detail to the caption for this figure.

6) Those two uses of "fixed-pattern noise" were in sentences regarding dark measurements, whereas "flat fielding" is generally thought of as a pixel-dependent gain adjustment applied to higher signal levels. The fixed-pattern noise can include flat-fielding variations as well as constant pixel-dependent offsets (say from zero) or bad pixels, both of which can have a larger relative contribution when measuring low signal levels such as the cases mentioned where the term "fixed-pattern" is used.

7) That really is Earth-reflected irradiance and not an uncertainty. While the plot shows the spectral dependence of various uncertainties, there is some (NASA-driven) desire to have a "simple" single-number uncertainty to quote rather than needing to show an entire spectral plot. Thus I computed a spectrally-averaged uncertainty weighted by a typical expected signal. Since these instrument uncertainties are relevant for Earth-looking scenes, I use a globally-averaged reflected-solar signal level (with spectral dependence shown in grey in the figure) and spectrally weighting the net uncertainty by it. This weighting means that spectral regions with very little signal have very little effect on the net average uncertainty; while the instrument uncertainties in regions of high spectral signal are more importantly weighted.

I've added some text (mostly naming of colors in the plots) to hopefully make this more clear in the paper; but as it is/was, I believe the wording is correct. (And hopefully it's both correct and more clear now. ;-)

Technical Correction Responses:

1) In the version you have, "SSI" should be defined on page 2, line 5 where I first used it. If not, it is defined around there on the latest version. Thanks!

2) With "NPOESS" now being a nearly-obsolete acronym from an obsolete program, this is probably even more important to define for future readers. Good idea, and I've done so now. (But my, what a *long* acronym "NPOESS" is — not to mention NPP!)

3) It is indeed missing, but that was intended to be implicit to avoid adding a fifth occurrence of "aperture" to the three sentences in that paragraph. I've instead now moved the first use of "apertures" in the subsequent sentence to this position, thinking that "apertures" may be more clearly implicit in that next sentence.

4) Opps, yep, definitely! I've got that corrected now.

Thanks for the meticulousness it takes to catch these kind of errors, and apologies for having missed them!

Best,

Greg

Please also note the supplement to this comment:
http://www.geosci-instrum-method-data-syst-discuss.net/gi-2016-37/gi-2016-37-AC3-supplement.pdf

---

## Referee Comment (RC3) · Anonymous Referee #2 · 6 Feb 2017

Thanks for your responses. I look forward to seeing the paper in print.